# R3DM: Enabling Role Discovery and Diversity Through Dynamics Models in Multi-agent Reinforcement Learning

**Harsh Goel** [1]  **Mohammad Omama** [1]  **Behdad Chalaki** [2]  **Vaishnav Tadiparthi** [2]  **Ehsan Moradi Pari** [2]
**Sandeep Chinchali** [1]

## Abstract

Multi-agent reinforcement learning (MARL) has achieved significant progress in large-scale traffic control, autonomous vehicles, and robotics. Drawing inspiration from biological systems where roles naturally emerge to enable coordination, role-based MARL methods have been proposed to enhance cooperation learning for complex tasks. However, existing methods exclusively derive roles from an agent's past experience during training, neglecting their influence on its future trajectories. This paper introduces a key insight: an agent's role should shape its future behavior to enable effective coordination. Hence, we propose Role Discovery and Diversity through Dynamics Models (R3DM), a novel role-based MARL framework that learns emergent roles by maximizing the mutual information between agents' roles, observed trajectories, and expected future behaviors. R3DM optimizes the proposed objective through contrastive learning on past trajectories to first derive intermediate roles that shape intrinsic rewards to promote diversity in future behaviors across different roles through a learned dynamics model. Benchmarking on SMAC and SMACv2 environments demonstrates that R3DM outperforms state-of-the-art MARL approaches, improving multi-agent coordination to increase win rates by up to 20%. The code is available at https://github.com/UTAustin-SwarmLab/R3DM.

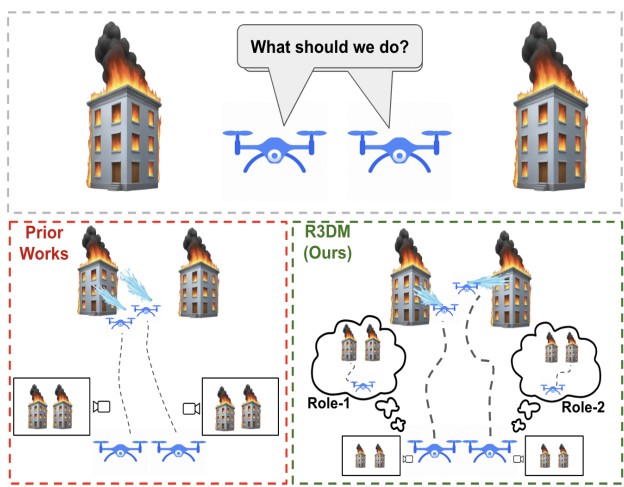

*Figure 1.* In a fire-fighting scenario with two drones, standard role-based multi-agent RL methods fail to distribute drones effectively, as roles are inferred from exhibited behavior. By linking roles to future expected behavior via a dynamics model, R3DM achieves better role differentiation and coordination.

## 1. Introduction

Multi-agent Reinforcement Learning (MARL) has seen increasing progress in board games (Meta Fundamental AI Research Diplomacy Team (FAIR) et al., 2022; Perolat et al., 2022), traffic signal control (Chu et al., 2019; Chinchali et al., 2018; Goel et al., 2023; Zhang et al., 2025), autonomous vehicles (Zhao et al., 2023; Han et al., 2023), stock markets (Bao & Liu, 2019), and collaborative robotics (Krnjaic et al., 2022; Zhang et al., 2022). A significant challenge is to learn policies for agents that enable effective coordination. Centralized Training Decentralized Execution (CTDE) is a common paradigm for training collaborative policies. CTDE methods such as QMIX (Rashid et al., 2020), QPlex (Wang et al., 2020a), MAPPO (Yu et al., 2022), MAAC (Iqbal & Sha, 2019) and VDN (Sunehag et al., 2018) commonly learn shared policy parameters from local observations across multiple agents. These approaches have been effective on benchmarks such as the Starcraft Multi-agent Challenge (SMAC) (Samvelyan et al., 2019; Ellis et al., 2024).

---

[1]Chandra Family Department of Electrical and Computer Engineering, The University of Texas at Austin, USA [2]Honda Research Institute, USA. Correspondence to: Harsh Goel <harshg99@utexas.edu>.

*Proceedings of the 42nd International Conference on Machine Learning*, Vancouver, Canada. PMLR 267, 2025. Copyright 2025 by the author(s).

However, these methods, which rely primarily on learning shared policy parameters for agents, often hinder the learning of diverse behaviors that individual agents need to exhibit to complete tasks reliably. To address this limitation, recent methods have explored various approaches: (i) encouraging individualized behaviors through diversity or skill-driven intrinsic rewards (Jiang & Lu, 2021a; Liu et al., 2023; Li et al., 2021; Liu et al., 2022), and (ii) explicitly training heterogeneous policies (Zhong et al., 2024). While these methods promote the learning of diverse behaviors, the focus on developing individualized behaviors often compromises effective coordination (Hu et al., 2024). Furthermore, sacrificing parameter sharing in heterogeneous setups (Liu et al., 2022; Zhong et al., 2024) reduces sample efficiency.

To balance diversity with efficient cooperation, role-based MARL methods like ACORM (Hu et al., 2024), CIA (Liu et al., 2023), RODE (Wang et al., 2021), GoMARL (Zang et al., 2023), and ROMA (Wang et al., 2020b) learn emergent roles from agents' past observations. These approaches encourage complementary and distinct behaviors within the parameter-sharing framework of CTDE training. However, deriving roles solely from past observations often fails to enforce effective coordination and cooperative behavior. For example, in a fire-fighting task involving drones extinguishing two separate fires (Fig. 1), if both drones have similar initial observations, they are likely to adopt identical roles early on. This leads to redundant behaviors, such as both drones moving toward the same fire, rather than distributing effectively to address both fires.

To address this issue, the central insight of this paper is that an agent's role should shape its future behavior, meaning an agent adopting different roles at any given moment will naturally follow distinct trajectories. For example, in a multi-drone firefighting scenario, a drone taking up a role to target any specific building would exhibit future observations and actions that naturally diverge as its targets change. This motivates our approach, **Role Discovery and Diversity through Dynamics Models** (R3DM), which maximizes the mutual information (MI) between agents' roles at a timestep, their observed trajectories up to that timestep, and expected future trajectories. By linking the current roles of agents to their future expected behaviors or trajectories, R3DM ensures agents learn more distinct yet complementary roles.

Our key contributions are summarized as follows:

1. We propose a novel information-theoretic objective between agents' roles, observed trajectories, and expected future trajectories. This objective is optimized jointly by a contrastive learning framework (Hu et al., 2024) for learning intermediate roles, which shapes intrinsic rewards devised from observation dynamics models to promote role-specific behavior.

2. We integrate this framework with an existing role-based MARL method, ACORM (Hu et al., 2024). By balancing task rewards with intrinsic rewards, our approach enhances the diversity of role-specific behavior, thereby enabling exploration.

3. We validate R3DM on SMAC (Samvelyan et al., 2019) and SMACv2 (Ellis et al., 2024) environments. Our method achieves superior coordination capabilities compared to existing role-based MARL approaches by showing up to 20% improvement in the final win rates in the challenging SMAC and SMACv2 environments such as 3s5z_vs_3s6z and protoss_10_vs_11.

## 2. Background

### 2.1. Preliminaries

Multi-agent tasks are modelled as a Decentralized Partially Observable Markov Decision Process (Dec-POMDP) $G = \langle I, S, O, A, P, R, \Omega, n, \gamma \rangle$, where $I = \{0, 1, \ldots, n-1\}$ is a finite set of $n$ agents, $s^t \in S$ is the global state from the continuous state space $S$. Each agent $i$ receives observation $o_i^t \in O_i$ where $O_i$ is its observation space, via the observation model $\Omega(s^t, i) : S \times I \to O_i$. This results in a joint observation space $O \equiv (O_1 \times O_2 \times \cdots \times O_n)$. Each agent selects an action $a_i^t \in A_i$ from its discrete action space $A_i$ by learning a policy $\pi_i : T_i^t \times A_i \to [0, 1]$, where $\tau_i^t \in (A_i \times O_i)^t \equiv T_i^t$ is the agent's observation-action history. This yields a joint action $\mathbf{a}^t = [a_1^t, \ldots, a_n^t] \in A$, where $A \equiv (A_1 \times A_2 \times \cdots \times A_n)$ is the global action space. The joint action policy is given by $\boldsymbol{\pi}$. The team transitions to a next state $s^{t+1}$ through a transition dynamics model $P(s^{t+1}|s^t, \mathbf{a}^t) : S \times A \times S \to [0, 1]$, and receives a reward $r^t = R(s^t, \mathbf{a}^t)$ shared by all agents. Here, the discount factor is $\gamma \in [0, 1]$. The joint policy induces the joint state action-value function $Q_{\text{tot}}^\pi(s, \mathbf{a}; \phi)$ at state $s$ and action $\mathbf{a}$ parameterized by a neural network $\phi$ as

$$Q_{\text{tot}}^\pi(s, \mathbf{a}; \phi) = \mathbb{E}_{a^{0:\infty}, s^{0:\infty} \sim \boldsymbol{\pi}} \left[ \sum_{j=0}^{\infty} \gamma^j r^j | s^0 = s, \mathbf{a}^0 = \mathbf{a} \right].$$
(1)

### 2.2. Value Function Factorization for Centralized Training with Decentralized Execution

MARL through the CTDE paradigm, such as QMIX (Rashid et al., 2020), has been a major focus in recent research due to the balance between centralized learning and decentralized decision-making. The key idea is the factorization of the value function, which composes the global action value function $Q_{\text{tot}}^\pi(s^t, \mathbf{a}^t; \phi) = f(Q_1(\tau_1^t, a_1^t; \phi_Q), \ldots, Q_n(\tau_n^t, a_n^t; \phi_Q), s^t, \mathbf{a}^t; \phi)$ through a function $f$, over individual utilities for each agent $Q_i(\tau_i^t, a_i^t; \phi_Q)$ with parameters $\phi_Q$. By adhering to the Individual-

Global-Maximum (IGM) principle, agents can independently maximize their utility functions to obtain an optimal global action, i.e., $\arg\max_{\mathbf{a}^t} Q^\pi_{\text{tot}}(s^t, \mathbf{a}^t; \phi) = [\arg\max_{a_1^t} Q_1(\tau_1^t, a_1^t; \phi_Q), ..., \arg\max_{a_n^t} Q_n(\tau_n^t, a_n^t; \phi_Q)]$.

### 2.3. Roles in MARL

Roles in MARL characterize distinct agent behaviors to foster coordination in CTDE frameworks with shared parameters. Given a Dec-POMDP $G = \langle I, S, O, A, P, R, \Omega, n, \gamma \rangle$, a role for agent $i$ is a time-varying latent variable $m_i^t \in M$ determined from its observation-action history $\tau_i^t \in (A_i \times O_i)^t$. Here, $M$ is a fixed set of roles with cardinality $|M|$. Knowing these roles apriori is infeasible unless they are handcrafted for the task, therefore, each agent learns a role-representation $z_i^t \in Z$ and a role-conditioned policy $\pi(\cdot|\tau_i^t, z_i^t) : T_i^t \times A_i \times Z \rightarrow [0,1]$ where $Z \subset \mathbb{R}^{d_r}$ is the role embedding space of dimension $d_r$. These roles emerge dynamically and enable agents to exhibit specialized yet complementary behaviors to coordinate for the given task.

## 3. Related Work

### 3.1. Role-Based Multi-Agent Reinforcement Learning

ROMA (Wang et al., 2020b) introduces a role embedding space based on agents' current observations, and observation-action histories, marking it as the first work to achieve this. Expanding on ROMA, RODE (Wang et al., 2021) and SIRD (Zeng et al., 2023) associate each role with a predefined subset of the joint action space. This approach simplifies the learning process by reducing action-space complexity. However, these methods maintain roles that access a static subset of actions during training, which prevents effective coordination. In contrast, LDSA (Yang et al., 2022), ALMA (Iqbal et al., 2022), and Xia et al. (2023) learn different sub-tasks within the MARL framework to dynamically assign roles to agents. Contrastive learning techniques have also been explored to learn role representations for improving multi-agent coordination. CIA (Liu et al., 2023) employs contrastive learning to learn roles by differentiating their credit or contributions to the team. Similarly, ACORM (Hu et al., 2024) leverages contrastive learning to learn roles that distinguish between different behavioral patterns. On the other hand, methods like SePS (Christianos et al., 2021) and GoMARL (Zang et al., 2023) promote heterogeneity and specialization by dynamically assigning agents into different subgroups, either using a dynamics model (SePS) or through weights on the individual utilities in the mixing network (GoMARL). Different from role-based methods that derive emergent roles for agents based on their observation-action histories, R3DM introduces an intrinsic reward to encourage agents to differentiate their expected future behaviors across different roles.

### 3.2. Intrinsic Rewards for Diversity in MARL

Diversity-based methods incentivize individualized or exploratory behaviors; for instance, MAVEN (Mahajan et al., 2019) is the first work that fosters diverse exploratory behaviors through latent space representations for hierarchical control. Additionally, EMC (Zheng et al., 2021) proposes to learn better behaviors through the means of an intrinsic reward derived from the error in the predictions of the Q-values. CDS (Li et al., 2021) promotes individuality by designing intrinsic rewards to maximize the mutual information between agent identities and roles and EOI (Jiang & Lu, 2021b) encourages agents to explore by training an observation-to-identity classifier, fostering individuality. In contrast to these methods, R3DM not only promotes individuality when beneficial, but also learns emergent role embeddings that distinguish between agents' past and expected future behaviors to enable a better balance between specialization and coordination in complex environments with varying team dynamics.

## 4. Method

In this section, we present our approach, R3DM, which enhances role learning in MARL through a novel information-theoretic objective that captures the relationship between an agent's role, past behavior, and future trajectory. In Section 4.1, we derive a tractable lower bound of this objective, decomposing it into two key components: (i) deriving intermediate role embeddings from agents' observation-action histories, and (ii) ensuring these embeddings guide diverse and distinct future behaviors to foster effective coordination. We maximize the first component using the contrastive learning framework introduced in (Hu et al., 2024) (Section 4.2), therefore obtaining intermediate role embeddings from agents' histories. Next, we introduce intrinsic rewards to maximize the second component (Section 4.3), ensuring that these embeddings generate diverse yet role-specific future behaviors. This reward balances diversity across trajectories associated with different roles while preserving the alignment of each trajectory with its corresponding role.

### 4.1. Role-based Diversity Objective

Given an agent $i$, its role $m_i^t$ and concatenated observation-action trajectory $\tau_i^{t+k}$ which comprises its observation-action history $\tau_i^t$ and future trajectory $\tau_i^{t+1:t+k}$ of $k$ steps, the MI objective is given as

$$I(\tau_i^{t+k}; m_i^t) = \mathbb{E}_{\tau_i^{t+k}, m_i^t}\left[\log\left(\frac{p(\tau_i^{t+k} \mid m_i^t)}{p(\tau_i^{t+k})}\right)\right]. \quad (2)$$

However, maximizing this objective is intractable in practice. Therefore, we utilize the following theorem to obtain a lower bound of this objective.

**Theorem 4.1.** *Given a set of roles $M$ with cardinality $|M|$, a role $m_i^t \in M$, and a concatenated observation-action trajectory $\tau_i^{t+k}$ which comprises its observation-action history $\tau_i^t$ and future trajectory $\tau_i^{t+1:t+k}$ with $k$ steps, if $e_i^t = f_{\theta_e}(\tau_i^t)$ denotes the embedding of the observation-action history obtained through a network $\theta_e$, and $z_i^t \sim f_{\theta_r}(z_i^t \mid e_i^t)$ denotes role embeddings obtained from a network $\theta_r$, then*

$$I(\tau_i^{t+k}; m_i^t) \geq \mathbb{E}_{e_i^t, z_i^t, m_i^t}\left[\log\left(\frac{p(z_i^t \mid e_i^t)}{p(z_i^t)}\right)\right] + I(\tau_i^{t+1:t+k}; z_i^t), \quad (3)$$

*where $I(\tau_i^{t+1:t+k}; z_i^t)$ is the MI between the future trajectory $\tau_i^{t+1:t+k}$ and role embedding $z_i^t$ at time $t$.*

We provide the proof in Appendix A.1. This theorem intuitively decomposes the objective into two parts i) learning role embeddings through the observation action history by maximizing the term $\mathbb{E}_{e_i^t, z_i^t, m_i^t}\left[\log\left(\frac{p(z_i^t \mid e_i^t)}{p(z_i^t)}\right)\right]$, and ii) maximizing the MI between the role embeddings and the future expected trajectory which results in intrinsic rewards as shown in Section 4.3. In the context of our firefighting example, drones would first derive intermediate role embeddings based on their current history. These embeddings act as compact representations of their roles, enabling the drones to differentiate their future trajectories, which would ultimately guide them to address fires in distinct sectors.

### 4.2. Role-embedding learning Objective

We maximize the term $\mathbb{E}_{e_i^t, m_i^t, z_i^t}\left[\log\left(\frac{p(z_i^t \mid e_i^t)}{p(z_i^t)}\right)\right]$ in Eq. 3 to learn intermediate role embeddings. However, maximizing this term directly is computationally intractable. To address this, we use the framework developed in previous work (ACORM (Hu et al., 2024)), which leverages contrastive learning to optimize the lower bound of this objective (Radford et al., 2021; Oord et al., 2018). For the sake of completeness, we explain the contrastive learning framework that clusters similar behaviors while separating distinct ones, effectively distinguishing between unique behavioral patterns and moving beyond simple agent identifiers as seen in (Li et al., 2021). For instance, in our firefighting scenario, drones assigned to extinguish fires in different buildings would be grouped into distinct roles based on their trajectories. Using this approach, we aim to derive intermediate role embeddings that capture meaningful distinctions in behavior. We utilize Theorem 4.2 to achieve this goal.

**Theorem 4.2.** *Let M denote a set of roles, and $m_i^t \in M$ denote a role. If $e_i^t = f_{\theta_e}(\tau_i^t)$ is an embedding of the observation-action history through a network $\theta_e$, and $z_i^t \sim$*

$f_{\theta_r}(z_i^t \mid e_i^t)$ *is a role-embedding from network $\theta_r$, then*

$$\mathbb{E}_{e_i^t, z_i^t, m_i^t} \log\left(\frac{p(z_i^t \mid e_i^t)}{p(z_i^t)}\right) \geq \log|M| +$$
$$\mathbb{E}_{e_i^t, z_i^t, m_i^t}\left[\log\frac{g(z_i^t, e_i^t)}{g(z_i^t, e_i^t) + \sum_{m_i^{t*} \in M/m_i^t} g(z_i^t, e_i^{t*})}\right], \quad (4)$$

*where $g(z_i^t, e_i^t)$ is a function whose optimal value is proportional to $\frac{p(z_i^t \mid e_i^t)}{p(z_i^t)}$, $m_i^{t*}$ is a role from $M$, and $e_i^{t*}$ is its corresponding observation-action history embedding.*

**Contrastive Learning.** Theorem 4.2 (proof provided in Appendix A.2) enables us to train the trajectory and role embedding networks, $\theta_e$ and $\theta_r$. First, agents are clustered into one of the $|M|$ roles based on their embeddings $e_i^t$, from which positive and negative pairs of role and agent embeddings are sampled. Following (Hu et al., 2024), we employ contrastive learning using bilinear products (Laskin et al., 2020) to formulate a score function $g(z_i^t, e_i^t)$ (from Theorem 4.2) that computes similarity between role and agent embeddings across clusters. We describe these in detail in the Appendix B.1.

### 4.3. Intrinsic Reward to optimize role based diversity

We introduce intrinsic rewards that maximizes the mutual information (MI) between role embeddings and future trajectories through Eq. 3 in Theorem 4.1. These intrinsic rewards are shown to enable diverse future behaviors while simultaneously enforcing role-specific future trajectories, thereby balancing exploration with role-aligned future behaviors. In our running example, these intrinsic rewards enforce the learning of role embeddings and subsequent policies that ensure that drones distribute effectively across the two fires. The intrinsic reward comprises: (i) *Policy Intrinsic Reward*, which encourages variability in an agent's policies through roles, and (ii) *Dynamics Intrinsic Reward*, which quantifies the predictive influence of role embeddings on an agent's future trajectory.

**Theorem 4.3.** *Given a local trajectory $\tau_i^t = \{o_0, a_0, o_1, a_1, \ldots, o_t\}$ and a learned role representation $z_i^t$ for an agent, the MI between the future trajectory $\tau_i^{t+1:t+k}$ and the role representation $z_i^t$ can be expressed as*

$$I(\tau_i^{t+1:t+k}; z_i^t) = \mathbb{E}_{\tau_i^t, z_i^t}\left[\sum_{l=t}^{t+k-1} \log\left(\frac{p(a_i^l \mid \tau_i^l, z_i^t)}{p(a_i^l \mid \tau_i^l)}\right) + \sum_{l=t}^{t+k-1} \log\left(\frac{p(o_i^{l+1} \mid \tau_i^l, z_i^t, a_i^l)}{p(o_i^{l+1} \mid \tau_i^l, a_i^l)}\right)\right], \quad (5)$$

*where $p(a_i^l \mid \tau_i^l, z_i^t)$ is the probability of taking action $a_i^l$ given the trajectory $\tau_i^l$ and role representation $z_i^t$, and*

$p(o_i^{l+1} \mid \tau_i^l, z_i^t, a_i^l)$ *is the probability of observing $o_i^{l+1}$ given the trajectory $\tau_i^l$, action $a_i^l$, and role representation $z_i^t$.*

The MI objective is decomposed into two terms: (i) the influence of the role on future action selection and (ii) the influence of the role on expected future observations. See Appendix A.3 for the proof.

**Policy Intrinsic Reward.** The diversity induced by the role embeddings in the policy of the agents is characterized by the term $\mathbb{E}_{\tau_i^t, z_i^t}\left[\sum_{l=t}^{t+k-1} \log\left(\frac{p(a_i^l \mid \tau_i^l, z_i^t)}{p(a_i^l \mid \tau_i^l)}\right)\right]$ in Eq. 5. To encourage this diversity, we devise a policy intrinsic reward. Since the policies in QMIX follow an $\epsilon$-greedy strategy, we obtain a lower bound for this ratio by leveraging the non-negativity of the KL divergence $\mathbb{D}_{KL}\left(p\left(\cdot \mid \tau_i^l, z_i^t\right) \| \text{SoftMax}\left(Q_i\left(\cdot \mid \tau_i^l, z_i^t; \phi_Q\right)\right)\right)$. Therefore, based on the equation

$$\mathbb{E}_{z_i^t, \tau_i^l, a_i^l}\left[\log\left(\frac{p(a_i^l \mid \tau_i^l, z_i^t)}{p(a_i^l \mid \tau_i^l)}\right)\right] \geq$$
$$\mathbb{E}_{z_i^t, \tau_i^l, a_i^l}\left[\log\left(\frac{\text{SoftMax}\left(Q_i\left(a_i^l \mid \tau_i^l, z_i^t; \phi_Q\right)\right)}{p(a_i^l \mid \tau_i^l)}\right)\right], \quad (6)$$

we devise the intrinsic rewards as

$$r_{i,\text{pol}}^t =$$
$$\sum_{l=t}^{t+k-1} \mathbb{D}_{KL}\left(\text{SoftMax}\left(Q_i\left(\cdot \mid \tau_i^l, z_i^t; \phi_Q\right)\right) \| p\left(\cdot \mid \tau_i^l\right)\right),$$
$$(7)$$

where $p\left(\cdot \mid \tau_i^l\right)$ is the average action probability over all role embeddings $z_i^t$, i.e., $p\left(\cdot \mid \tau_i^l\right) = \mathbb{E}_{z_i^t}\left[\text{SoftMax}\left(Q_i\left(\cdot \mid \tau_i^l, z_i^t; \phi_Q\right)\right)\right]$. Note that $\mathbb{E}_{z_i^t}$ is taken over role embeddings for all agents $i \in I$ at timestep $t$, and $\mathbb{D}_{KL}$ denotes the KL divergence. This intrinsic reward encourages diverse policies to be aligned with roles. We provide a detailed explanation in Appendix B.5.

**Dynamics Intrinsic Reward.** We derive the dynamics intrinsic reward from the latter term in Eq. 5 $\mathbb{E}_{\tau_i^t, z_i^t}\left[\sum_{l=t}^{t+k-1} \log\left(\frac{p(o_i^{l+1} \mid \tau_i^l, z_i^t, a_i^l)}{p(o_i^{l+1} \mid \tau_i^l, a_i^l)}\right)\right]$, which is equivalent to the mutual information between the future predicted observations $o_i^{t+1:t+k}$ and the embedding role $z_i^t$. To compute this ratio we learn two models: (i) a role-agnostic model that approximates observation dynamics $p\left(o_i^{l+1} \mid \tau_i^l, a_i^l\right)$ (ii) a role-conditioned dynamics model $q_\psi\left(o_i^{l+1} \mid \tau_i^l, z_i^t, a_i^l\right)$ to approximate the posterior observation dynamics $p\left(o_i^{l+1} \mid \tau_i^l, z_i^t, a_i^l\right)$. We then obtain a lower bound due to the non-negativity of the KL Divergence $\mathbb{D}_{KL}(q_\psi\left(o_i^{l+1} \mid \tau_i^l, z_i^t, a_i^l\right) \| p\left(o_i^{l+1} \mid \tau_i^l, z_i^t, a_i^l\right))$ (see Ap-

pendix B.5) through ELBO (Jordan et al., 1999) as:

$$\mathbb{E}_{z_i^t, \tau_i^l, a_i^l}\left[\log\left(\frac{p(o_i^{l+1} \mid \tau_i^l, z_i^t, a_i^l)}{p(o_i^{l+1} \mid \tau_i^l, a_i^l)}\right)\right]$$
$$\geq \mathbb{E}_{z_i^t, \tau_i^l, a_i^l}\left[\log\left(\frac{q_\psi(o_i^{l+1} \mid \tau_i^l, z_i^t, a_i^l)}{p(o_i^{l+1} \mid \tau_i^l, a_i^l)}\right)\right]. \quad (8)$$

We leverage the Recurrent State Space Model (RSSM) from the Dreamer series (Hafner et al., 2019; 2023) for the observation dynamics $q_\psi(o_i^{l+1} \mid \tau_i^l, z_i^t, a_i^l)$ and $p(o_i^{l+1} \mid \tau_i^l, z_i^t, a_i^l)$. The role conditioned variant of RSSM consists of the following components: **1)** A sequence model $q_{\psi_{\text{seq}}}(h_i^t \mid \tau_i^{t-1})$ with parameters $\psi_{\text{seq}}$, encoding the trajectory $\tau_i^{t-1}$ into a hidden representation $h_i^t$, **2)** An observation encoder $q_{\psi_e}(d_i^t \mid h_i^t, o_i^t)$ with parameters $\psi_e$, mapping observations $o_i^t$ into a latent representation $d_i^t$, **3)** A dynamics predictor $q_{\psi_{\text{dyn}}}(d_i^t \mid h_i^t, z_i^t)$ with parameters $\psi_{\text{dyn}}$, predicting the latent representation $d_i^t$ from $h_i^t$, and **4)** An observation decoder $q_{\psi_{\text{dec}}}(o_i^t \mid h_i^t, d_i^t, z_i^t)$ with parameters $\psi_{\text{dec}}$, reconstructing observations $o_i^t$. As stated earlier, we learn two models: a role-agnostic RSSM model $p(o_i^{l+1} \mid \tau_i^l, a_i^l)$ and a role-conditioned RSSM model whose corresponding sequence model $q_\psi(o_i^{l+1} \mid \tau_i^l, z_i^t, a_i^l)$. These models defer in the implementation of the dynamics predictor and observation decoder sub-networks, where for the role-agnostic model these are $p_{\phi_{\text{dyn}}}(d_i^t \mid h_i^t)$ with parameters $\phi_{\text{dyn}}$ and $p_{\phi_{\text{dec}}}(o_i^t \mid h_i^t, d_i^t)$ with parameters $\phi_{\text{dec}}$ respectively. This model comprises a separate sequence model $p_{\phi_{\text{seq}}}(h_i^t \mid \tau_i^{t-1})$ with parameters $\phi_{\text{seq}}$ and an encoder model $p_{\phi_{\text{enc}}}(h_i^t \mid \tau_i^{t-1})$ with parameters $\phi_{\text{enc}}$. Finally, the dynamics intrinsic reward that captures the influence of $z_i^t$ on predicting the next observation is given by (see Appendix B.5 for derivation)

$$r_{\text{i,dyn}}^t = \sum_{l=t}^{t+k-1}\Big(\beta_1\Big[\log q_{\psi_{\text{dec}}}\big(o_i^{l+1} \mid h_i^{l+1}, d_i^{l+1}, z_i^t\big)$$
$$+ \beta_2 \log q_{\psi_{\text{dyn}}}\big(d_i^{l+1} \mid h_i^{l+1}, z_i^t\big)\Big]$$
$$- \Big[\log p_{\phi_{\text{dec}}}\big(o_i^{l+1} \mid h_i^{l+1}, d_i^{l+1}\big)$$
$$+ \beta_2 \log p_{\phi_{\text{dyn}}}\big(d_i^{l+1} \mid h_i^{l+1}\big)\Big]\Big), \quad (9)$$

where $\beta_1$ and $\beta_2$ are hyperparameters that balance the decoder and latent-dynamics terms. The first bracket captures the log-likelihood terms of the role-conditioned DreamerV3 model, and the second bracket captures the same terms under the role-agnostic model. Note that the hyperparameter $\beta_1$ trades off between the diversity of trajectories across roles and the specificity of a trajectory to a given role. This incentivizes roles that yield distinct behaviors.

### 4.4. Final Learning Objective

Given the task reward $r^t$ at timestep $t$, we compute the total intrinsic reward $r_{\text{int}}^t = \sum_{i \in I} \beta_3 r_{i,\text{pol}}^t + r_{i,\text{dyn}}^t$ summed over all agents. Note that $\beta_3$ is a hyperparameter to weigh the policy intrinsic reward relative to the dynamics intrinsic reward. Therefore, the final centralized Q learning objective is

$$\mathcal{L}_{TD}(\theta) = [r^t + \alpha r_{\text{int}}^t + \gamma \max_{a^{t+1}} Q_{\text{tot}}(s^{t+1}, a^{t+1}; \phi^-)$$
$$- Q_{\text{tot}}(s^t, a^t; \phi)]^2. \quad (10)$$

Here, $\gamma$ is the discount factor, $\phi^-$ are the parameters of the frozen target mixing Q network, and $\alpha$ is a hyperparameter that weighs the intrinsic reward with respect to task reward. We detail the choice of the hyper-parameters in Appendix B and the entire algorithm in Appendix C.

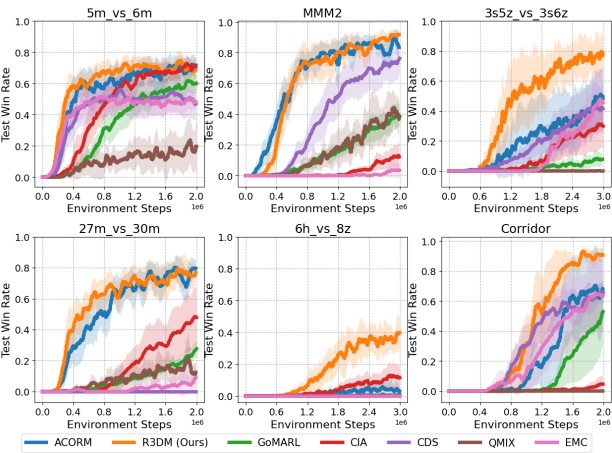

*Figure 2.* **Test Win Rate of R3DM compared to baselines on 6 maps in the SMAC.** We observe that R3DM improves sample efficiency, and converges to higher win rates on super-hard environments such as 3s5z_vs_3s6z, Corridor, and 6h_vs_8z.

## 5. Results

In this section, we present the results of our experiments conducted on the challenging environments on both the Starcraft Multi-Agent Challenge (SMAC) benchmarks - SMAC and SMACv2. Our primary goal is to evaluate the performance of R3DM by addressing the following questions.

1. Does R3DM facilitate the learning of winning coordination strategies in multi-agent domains (Sections 5.1 and 5.2) through intrinsic rewards?

2. Does R3DM learn roles that qualitatively show distinct future behavior to enable cooperation (Section 5.3)?

3. What design or hyperparameter choices are crucial to R3DM ? (Section 5.4)

**Environments.** The experiments are conducted on two benchmarks: SMAC and SMACv2, which comprise several micro-management scenarios to control each unit with limited local observations, to defeat an enemy team controlled by a built-in game AI. SMAC includes various scenarios with differing terrain layouts and unit types with varying difficulty levels. We focus on the 6 hard and super-hard scenarios – 5m_vs_6m, MMM2, 3s5z_vs_3s6z, 27m_vs_30m, 6h_vs_8z, and Corridor – that require more skillful coordination. In contrast to SMAC, SMAC-v2 introduces stochasticity in the initialization conditions that further challenge agents to explore diverse behaviors that foster skillful coordination. All experiments are conducted using SMAC version SC 2.4.10, and we note that performance comparisons do not translate across different SMAC versions.

**Baselines.** In addition to QMIX, we compare the performance of R3DM against five state-of-the-art baselines, that learn roles through contrastive learning (ACORM & CIA), learn grouping mechanism for agents (GoMARL), and devise intrinsic rewards for diversity (CDS) and exploration (EMC), on the same 5 random seeds.

### 5.1. Results on SMAC

R3DM demonstrates significant improvements over baseline methods, particularly in challenging SMAC scenarios such as 6h_vs_8z, Corridor, and 3s5z_vs_3s6z, where agents face stronger enemy teams. These results highlight the effectiveness of our approach in learning cooperative policies as well as achieving higher sample efficiency. For example, in Corridor and 3s5z_vs_3s6z, our method outperforms others by achieving faster convergence and higher test win rates compared to other baselines.

We also note that R3DM outperforms ACORM, on which our method is based, in many environments. This is likely due to the limitation in ACORM, which restricts the exploration of cooperative strategies due to the strong interdependence between agent identities and past trajectories. In contrast, R3DM not only promotes additional diversity through intrinsic rewards but also fosters role-specific future behavior. Therefore, we exhibit improved sample efficiency through exploration and better performance due to the discovery of better-coordinated behaviors amongst agents.

### 5.2. Results on SMACv2

The stochasticity introduced in SMACv2 such as variations in agent types, starting positions, and configurations, challenges MARL algorithms to explore behaviors that generalize well. In this context, R3DM demonstrates superior performance over established baselines in terms of test win rates and test cumulative reward. We note that R3DM outperforms Vanilla QMIX due to the role learning mechanism

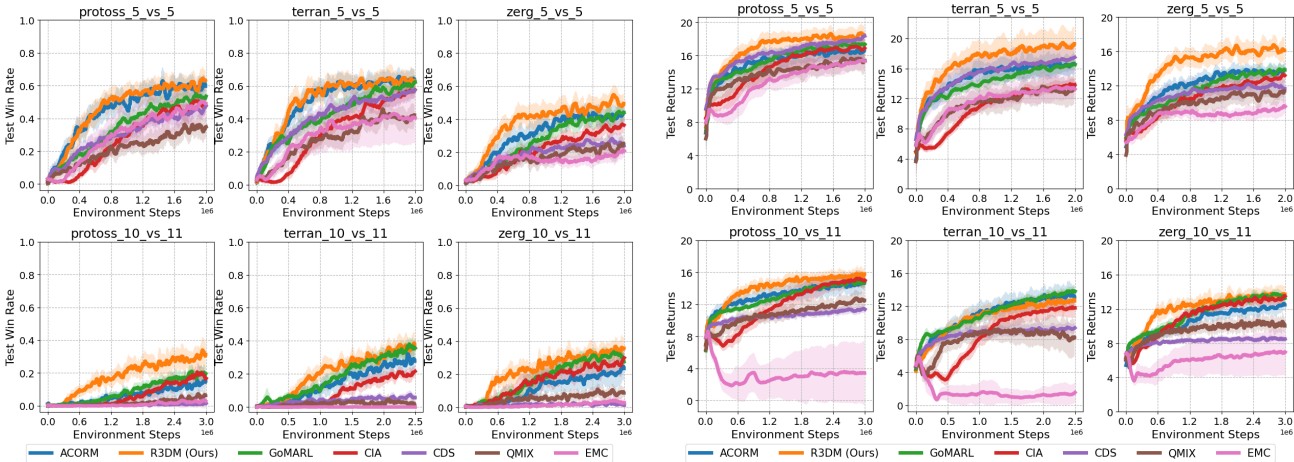

*Figure 3.* **Comparison of Test Win Rate and Test Cumulative Reward on the SMACv2 suite of environments.** We observe that R3DM showcases better returns highlighting better strategies learned in environments such as protoss_5_vs_5 and terran_5_vs_5, where its test win rate is equivalent to the best-performing baseline ACORM. In zerg_5_vs_5, protoss_10_vs_11 and zerg_10_vs_11 environments, R3DM outperforms the baselines in terms of test win rates. Note that we report the means (solid line) and standard deviation (shaded regions) across 5 seeds.

that results in diverse behaviors. Furthermore, R3DM outperforms intrinsic reward methods such as CDS and EMC. CDS incentivizes diversity by ensuring distinct trajectories across all agents, therefore, this leads to the learning of fragmented behaviors ill-suited for skillful coordination. This is evident from the mean test cumulative reward plots, which are comparatively poorer across all environments. Additionally, we notice that EMC does not make good learning progress, especially on the asymmetric 10_vs_11 environments, primarily due to its emphasis on exploration.

Finally, we discuss the performance of R3DM against Go-MARL, and role-based MARL methods CIA and ACORM. In the 5_vs_5 suite of environments, while R3DM exhibits similar win rates compared to ACORM we observe that it outperforms most baselines in test cumulative rewards. This indicates that R3DM learns more efficient winning strategies. In the more challenging assymetric 10_vs_11 environments, we observe that all learning algorithms fail to make sufficient learning progress due to the inherent stochasticity that results in a wider range of unit compositions. We observe that R3DM is marginally more sample efficient and exhibits improved win rates and rewards in the protoss and zerg-based environments.

### 5.3. Qualitative Analysis

We conduct a qualitative analysis of the strategies learned by our method in contrast to the best-performing baseline ACORM to depict the behavioral differences between our method and ACORM. We plot the visualization of the learned policy in the 3s5z_vs_3s6z environment and the TSNE plots of learned role representations at timesteps **1, 20, 35, and 50** in Fig. 4. Initial observations at $t = 1$ reveal

distinct role embedding clusters in both methods, indicating proper role initialization. However, by $t = 20$, ACORM's role-conditioned policies show reduced diversity, with all agent clusters converging to attack enemy units en masse. In contrast, R3DM demonstrates strategic differentiation: a green-coded stalker agent (highlighted in red) diverts three enemy zealots from the main group, while the remaining agents split into two specialized subteams to eliminate the weakened team. This implies that R3DM learns role embeddings that enable the future behavioral diversity needed for coordination. Subsequently, we observe that the agents controlled by policies learned by ACORM are losing to the enemy agents as the episode progresses to $t = 35$ and $t = 50$. On the other hand, since R3DM learned roles that enabled more distinct behavior earlier in the episode, agents effectively defeat enemy agents. We believe that this is observed due to the introduction of intrinsic rewards that optimize for the specificity of the trajectory to a role, while also enabling exploration by increasing the entropy of the future behaviors demonstrated across all identified roles.

### 5.4. Ablation

We conduct 3 ablations on the design choices - 1) the impact of the horizon of the imagined trajectories for reward computation, 2) cardinality of the number of roles, and 3) impact of contrastive learning.

**Impact of Imagination Horizon for Rewards.** We analyze the effect of varying the number of imagination steps used to generate role-conditioned future trajectories with $k$ timesteps for intrinsic reward computation (note that these are equivalent). In R3DM, a single imagination step is employed, as increasing this to two steps yields no statisti-

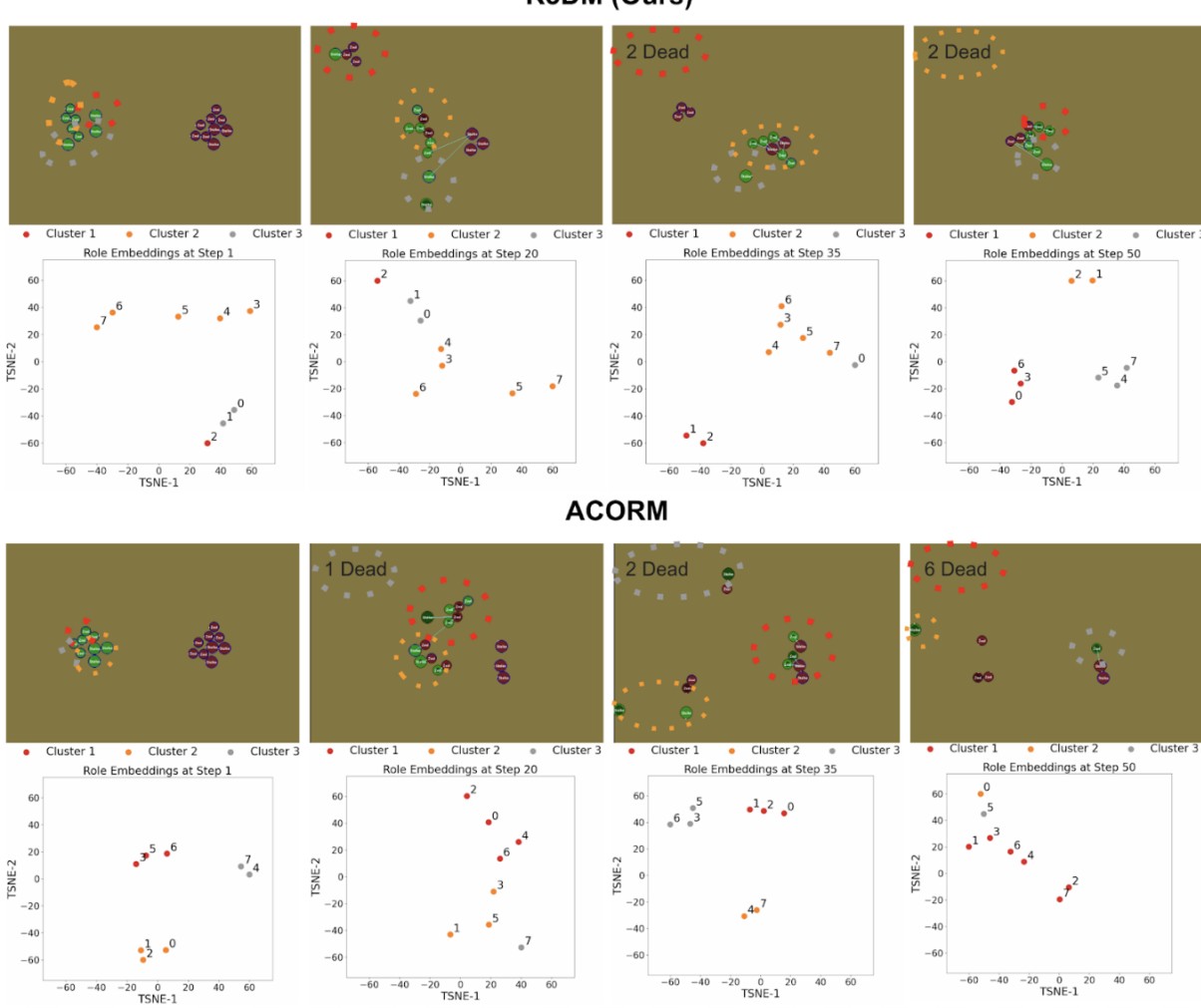

*Figure 4.* **We show qualitative results on the 3s_vs_5z environment with the corresponding role embeddings and the clusters.** R3DM learns a better strategy compared to the baseline ACORM, where one stalker agent, as shown in timestep 20, successfully learns a distinct role that lures enemy zealots for the main team to beat a weakened enemy force in the subsequent timesteps. While ACORM learns differentiated roles based on past observations, the resulting policies are inadequate to win against the enemy team.

cally significant performance improvement. Extending the horizon further (e.g., 5 or 10 steps) results in performance degradation, with the most pronounced decline occurring at 10 steps. We hypothesize that this degradation stems from compounding prediction errors in the model, which is conditioned solely on an agent's local observation-action history. These errors propagate over successive imagination steps, producing intrinsic rewards with increased variance and bias that destabilize policy optimization.

**Impact of Role Cardinality.** We evaluate R3DM with the number of role clusters $|M|$ ranging from 2 to 8 in the 3s5z_vs_3s6z environment. While the final performance remains statistically indistinguishable across configurations, training with 3 role clusters exhibits superior sample efficiency compared to higher cardinalities. This suggests that excessive role specialization (e.g., 8 clusters) introduces heterogeneity, whereas a moderate number (3 clusters) optimally balances coordination and specialization.

**Impact of Contrastive Learning.** To isolate the contributions of contrastive learning (CL) and intrinsic rewards, we compare three variants: 1) R3DM without intrinsic rewards (equivalent to ACORM), 2) R3DM with intrinsic rewards but without CL, and 3) Full R3DM (intrinsic rewards + CL). The variant without CL underperforms R3DM but still surpasses ACORM, demonstrating that intrinsic rewards, which are designed to maximize trajectory entropy conditioned on roles, drive significant performance gains. CL further enhances results by enforcing distinct role representations, reducing conditional entropy between trajectories and role assignments. This highlights the complementary impact of intrinsic rewards (exploration) and CL (representation disentanglement) in the framework.

*Figure 5.* **We conduct an ablation study on the 3s5z_vs_3s6z environment to evaluate the impact of: 1) Imagination Horizon for Reward, 2) Number of Roles, and 3) Role optimization without Contrastive Learning.** We observe that a R3DM with (a) shorter imagination horizons ($k = 1, 2$) outperform longer ones ($k = 5, 10$) due to reduced compounding errors, (b) moderate role cardinality ($N_r = 3$) achieves faster convergence despite similar final performance across configurations, and (c) full R3DM with both contrastive learning and intrinsic rewards demonstrates superior performance compared to the partial implementations.

## 6. Limitations and Future Work

While R3DM presents a key insight to improve role-based MARL, it has a few limitations. First, the number of roles in the environment needs to be set apriori, therefore, making it a hyperparameter. Future work can explore removing this as a requirement and instead dynamically derive roles from the replay buffer. Additionally, R3DM computes intrinsic rewards through a world model on the local observation dynamics that do not take into account the influence of other agents' actions or roles on the observations of the ego agent. Future work can be extended to incorporate more sophisticated dynamics models that would more accurately compute the influence of an ego-agent's role embedding on its future trajectory to learn better intrinsic rewards that would boost sample efficiency.

## 7. Conclusion and Future Work

To overcome the limitations of existing role-based MARL methods in fostering effective cooperation, we propose Role Discovery and Diversity through Dynamics Models (R3DM), a novel role-based MARL framework. R3DM introduces a mutual information-based objective that establishes a direct connection between agents' roles, their observed trajectories, and their expected future behaviors. This enables the emergence of specialized and diverse behaviors while balancing exploration and role-specific specialization. R3DM optimizes the proposed objective through contrastive learning on past trajectories to first derive intermediate roles that subsequently shape intrinsic rewards to promote diversity in future behaviors across different roles through a learned dynamics model. Experimental evaluations on challenging benchmarks such as SMAC and SMACv2, show improvements in coordination capabilities, therefore improving test win rates and cumulative rewards. These results highlight the potential of our framework and pave the way to incorporate model-based RL into MARL algorithms.

## Impact Statement

Our work presents a new algorithm to advance the field of Multi-agent Reinforcement Learning. Beyond its potential impact on multi-player games and environments, we believe that the implications of our work would have minimal societal consequences.

## Acknowledgement

We would like to thank Po-han Li and Jiaxun Cui for their helpful feedback on improving the paper. This work was supported in part by the Honda Research Institute, National Science Foundation Grants No. 2133481 and No. 2148186, and Cisco Systems, Inc., under MRA MAG00000005, through the 6G@UT center within the Wireless Networking and Communications Group (WNCG) at the University of Texas at Austin. Any opinions, findings, and conclusions or recommendations expressed in this material are those of the authors and do not necessarily reflect the views of the National Science Foundation.

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

## A. Proofs

### A.1. Theorem 4.1

**Theorem 4.1** Given a set of roles $M$ with cardinality $|M|$, a role $m_i^t \in M$, and a concatenated observation-action trajectory $\tau_i^{t+k}$ which comprises its observation-action history $\tau_i^t$ and future trajectory $\tau_i^{t+1:t+k}$ with $k$ steps, if $e_i^t = f_{\theta_e}(\tau_i^t)$ denotes the embedding of the observation-action history obtained through a network $\theta_e$, and $z_i^t \sim f_{\theta_r}(z_i^t \mid e_i^t)$ denotes role embeddings obtained from a network $\theta_r$, then

$$I(\tau_i^{t+k}; m_i^t) \geq \mathbb{E}_{e_i^t, z_i^t, m_i^t} \left[ \log \left( \frac{p(z_i^t \mid e_i^t)}{p(z_i^t)} \right) \right] + I(\tau_i^{t+1:t+k}; z_i^t), \tag{11}$$

where $I(\tau_i^{t+1:t+k}; z_i^t)$ is the MI between the future trajectory $\tau_i^{t+1:t+k}$ and role embedding $z_i^t$ at time $t$.

*Proof.* We begin with the mutual information objective

$$I(\tau_i^{t+k}; m_i^t) = \mathbb{E}_{\tau_i^{t+k}, m_i^t} \left[ \log \frac{p(\tau_i^{t+k} | m_i^t)}{p(\tau_i^{t+k})} \right], \tag{12}$$

where the trajectory $\tau_i^{t+k}$ is a concatenation of the observation-action history and future trajectory of an agent $i$. Maximizing this objective is intractable as the role $m_i^t$ is unknown, therefore, we obtain a lower bound of this objective by learning a role encoder $\theta_r$ to obtain an intermediate role representation $z_i^t$ from embeddings of an agent's observation-action history $e_i^t$ through an encoder $\theta_e$.

We then make the following simplifications to the ratio $\frac{p(\tau_i^{t+k} | m_i^t)}{p(\tau_i^{t+k})}$:

$$
\begin{aligned}
\frac{p(\tau_i^{t+k} | m_i^t)}{p(\tau_i^{t+k})} &= \frac{\int_{e_i^t, z_i^t, \tau_i^t} p(\tau_i^{t+k} \mid z_i^t, \tau_i^t) p(z_i^t \mid e_i^t) p(e_i^t \mid \tau_i^t, m_i^t) p(\tau_i^t) de_i^t dz_i^t d\tau_i^t}{\int_{\tau_i^t} p(\tau_i^{t+k} \mid \tau_i^t) p(\tau_i^t) d\tau_i^t} \\
&\approx \mathbb{E}_{e_i^t, z_i^t, m_i^t} \frac{p(\tau_i^{t+k} \mid z_i^t, \tau_i^t) p(z_i^t \mid e_i^t)}{p(\tau_i^{t+k} \mid \tau_i^t) p(z_i^t)} \qquad \text{(We assume that } p(e_i^t \mid \tau_i^t, m_i^t) \approx p(e_i^t | m_i^t)) \\
&= \mathbb{E}_{e_i^t, z_i^t, m_i^t} \frac{p(\tau_i^{t+1:t+k} \mid z_i^t) p(z_i^t \mid e_i^t)}{p(\tau_i^{t+1:t+k}) p(z_i^t)},
\end{aligned}
$$

where $\tau_i^{t+1:t+k}$ denotes the $k$ steps of the agent's future trajectory that comprises future observations and future actions.

Therefore, the mutual information introduced in Eq. 12 between $\tau_i^{t+k}$ and role $m_i^t$ is

$$I(\tau_i^{t+k}; m_i^t) \approx \mathbb{E}_{\tau_i^{t+k}, m_i^t} \left[ \log \left( \mathbb{E}_{e_i^t, z_i^t, m_i^t} \frac{p(\tau_i^{t+1:t+k} \mid z_i^t) p(z_i^t \mid e_i^t)}{p(\tau_i^{t+1:t+k}) p(z_i^t)} \right) \right].$$

Leveraging Jensen's inequality, we obtain a lower bound for this mutual information as follows:

$$
\begin{aligned}
I(\tau_i^{t+k}; m_i^t) &\approx \mathbb{E}_{\tau_i^{t+k}, m_i^t} \left[ \log \left( \mathbb{E}_{e_i^t, z_i^t, m_i^t} \frac{p(\tau_i^{t+1:t+k} \mid z_i^t) p(z_i^t \mid e_i^t)}{p(\tau_i^{t+1:t+k}) p(z_i^t)} \right) \right] \\
&\geq \mathbb{E}_{\tau_i^{t+k}, m_i^t, e_i^t, z_i^t} \left[ \log \frac{p(\tau_i^{t+1:t+k} \mid z_i^t) p(z_i^t \mid e_i^t)}{p(\tau_i^{t+1:t+k}) p(z_i^t)} \right] \\
&= \mathbb{E}_{\tau_i^{t+k}, z_i^t, m_i^t} \left[ \log \frac{p(\tau_i^{t+1:t+k} \mid z_i^t)}{p(\tau_i^{t+1:t+k})} \right] + \mathbb{E}_{z_i^t, e_i^t, m_i^t} \left[ \log \frac{p(z_i^t \mid e_i^t)}{p(z_i^t)} \right] \\
&= I(\tau_i^{t+1:t+k}; z_i^t) + \mathbb{E}_{z_i^t, e_i^t, m_i^t} \left[ \log \frac{p(z_i^t \mid e_i^t)}{p(z_i^t)} \right].
\end{aligned}
$$

This completes the proof, where we obtain a tractable lower bound of the original MI objective conditioned on the unknown role variable $m_i^t$. By optimizing this lower bound, we can optimize the original objective.

$\square$

### A.2. Theorem 4.2

**Theorem 4.2**

Let M denote a set of roles, and $m_i^t \in M$ denote a role. If $e_i^t = f_{\theta_e}(\tau_i^t)$ is an embedding of the observation-action history through a network $\theta_e$, and $z_i^t \sim f_{\theta_r}(z_i^t \mid e_i^t)$ is a role-embedding from network $\theta_r$, then

$$\mathbb{E}_{e_i^t, z_i^t, m_i^t} \log \left( \frac{p(z_i^t \mid e_i^t)}{p(z_i^t)} \right) \geq \log |M| + \mathbb{E}_{e_i^t, z_i^t, m_i^t} \left[ \log \frac{g(z_i^t, e_i^t)}{g(z_i^t, e_i^t) + \sum_{m_i^{t*} \in M/m_i^t} g(z_i^t, e_i^{t*})} \right],$$

where $g(z_i^t, e_i^t)$ is a function whose optimal value is proportional to $\frac{p(z_i^t \mid e_i^t)}{p(z_i^t)}$, $m_i^{t*}$ is a role from $M$, and $e_i^{t*}$ is its corresponding observation-action history embedding.

*Proof.* To prove this theorem, we need to first show that the optimal value of the score function $g(z_i^t, e_i^t)$ is proportional to the ratio $\frac{p(z_i^t \mid e_i^t)}{p(z_i^t)}$ which describes the ratio between the conditional likelihood of obtaining the role embedding from the agent embedding and the marginal likelihood of the role embeddings.

Therefore, we begin the proof by proving the following lemma.

**Lemma A.1.** *The value of the function $g(z_i^t, e_i^t)$ that minimizes the following term*

$$-\mathbb{E}_{e_i^t, z_i^t, m_i^t} \left[ \log \frac{g(z_i^t, e_i^t)}{g(z_i^t, e_i^t) + \sum_{m_i^{t*} \in M/m_i^t} g(z_i^t, e_i^{t*})} \right] \tag{13}$$

*is proportional to $\frac{p(z_i^t \mid e_i^t)}{p(z_i^t)}$, where $m_i^{t*}$ is a role from $M$ and $e_i^{t*}$ is its corresponding observation-action history embedding.*

*Proof.* Following the proof in (Oord et al., 2018), it can be seen that the function in Eq. 13 is akin to the categorical loss of a model or a function $g(z_i^t, e_i^t)$. Therefore, $g(z_i^t, e_i^t)$ is the optimal probability $p(i = m_i^t \mid e_i^t, Z)$ that the agent $i$ takes up the role $m_i^t$ based on the agent embedding $e_i^t$ and the set of all role embeddings $Z = \{z_i^t \mid i \in I\}$ for all agents in $I$.

We can write the optimal probability $p(i = m_i^t \mid e_i^t, Z)$ using Bayes Theorem as:

$$\begin{aligned}
p(i = m_i^t \mid e_i^t, Z) &= \frac{p(Z \mid m_i^t, e_i^t)}{\sum_{m_i^t} p(Z \mid m_i^t, e_i^t)} \\
&= \frac{\prod_{j \in I/i} \left( p(z_j^t) \right) p(z_i^t \mid e_i^t)}{\sum_{m_i^t} \prod_{j \in I/i} \left( p(z_j^t) \right) p(z_i^t \mid e_i^t)} \qquad \text{(Since } p(z_i^t \mid m_i^t, e_i^t) = p(z_i^t \mid e_i^t)) \\
&= \frac{\frac{p(z_i^t \mid e_i^t)}{p(z_i^t)}}{\sum_{m_i^t} \frac{p(z_i^t \mid e_i^t)}{p(z_i^t)}} \\
&\propto \frac{p(z_i^t \mid e_i^t)}{p(z_i^t)}.
\end{aligned}$$

Since the function $g(z_i^t, e_i^t)$ is equivalent to the optimal probability $p(i = m_i^t \mid e_i^t, Z)$, it is proportional to $\frac{p(z_i^t \mid e_i^t)}{p(z_i^t)}$. This concludes the proof of the Lemma. $\square$

Substituting the optimal value of the score function in Eq. 13, we get

$$= -\mathbb{E}_{e_i^t, z_i^t, m_i^t} \left[ \log \frac{\frac{p(z_i^t | e_i^t)}{p(z_i^t)}}{\frac{p(z_i^t | e_i^t)}{p(z_i^t)} + \sum_{m_i^{t*} \in M/m_i^t} \frac{p(z_i^t | e_i^{t*})}{p(z_i^t)}} \right]$$

$$= \mathbb{E}_{e_i^t, z_i^t, m_i^t} \left[ 1 + \frac{p(z_i^t)}{p(z_i^t | e_i^t)} \sum_{m_i^{t*} \in M/m_i^t} \frac{p(z_i^t | e_i^{t*})}{p(z_i^t)} \right]$$

$$\approx \mathbb{E}_{e_i^t, z_i^t, m_i^t} \left[ \log \left( 1 + (|M| - 1) \frac{p(z_i^t)}{p(z_i^t | e_i^t)} \mathbb{E}_{m_i^{t*} \in M/m_i^t} \frac{p(z_i^t | e_i^{t*})}{p(z_i^t)} \right) \right]$$

$$\approx \mathbb{E}_{e_i^t, z_i^t, m_i^t} \left[ \log \left( 1 + (|M| - 1) \frac{p(z_i^t)}{p(z_i^t | e_i^t)} \right] \right)$$

$$\geq \mathbb{E}_{e_i^t, z_i^t, m_i^t} \left[ \log \left( |M| \frac{p(z_i^t)}{p(z_i^t | e_i^t)} \right] \right)$$

$$= \log |M| - \mathbb{E}_{e_i^t, z_i^t, m_i^t} \left[ \log \frac{p(z_i^t | e_i^t)}{p(z_i^t)} \right].$$

Therefore, we get the expression

$$-\mathbb{E}_{e_i^t, z_i^t, m_i^t} \left[ \log \frac{g(z_i^t, e_i^t)}{g(z_i^t, e_i^t) + \sum_{m_i^{t*} \in M/m_i^t} g(z_i^t, e_i^{t*})} \right] \geq \log |M| - \mathbb{E}_{e_i^t, z_i^t, m_i^t} \left[ \log \frac{p(z_i^t | e_i^t)}{p(z_i^t)} \right].$$

or

$$\mathbb{E}_{e_i^t, z_i^t, m_i^t} \left[ \log \frac{p(z_i^t | e_i^t)}{p(z_i^t)} \right] \geq \log |M| + \mathbb{E}_{e_i^t, z_i^t, m_i^t} \left[ \log \frac{g(z_i^t, e_i^t)}{g(z_i^t, e_i^t) + \sum_{m_i^{t*} \in M/m_i^t} g(z_i^t, e_i^{t*})} \right]$$

This concludes the proof. $\square$

### A.3. Theorem 4.3

**Theorem 4.3** Given a local trajectory $\tau_i^t = \{o_0, a_0, o_1, a_1, \ldots, o_t\}$ and a learned role representation $z_i^t$ for an agent, the MI between the future trajectory $\tau_i^{t+1:t+k}$ and the role representation $z_i^t$ can be expressed as

$$I(\tau_i^{t+1:t+k}; z_i^t) = \mathbb{E}_{\tau_i^t, z_i^t} \left[ \sum_{l=t}^{t+k-1} \log \left( \frac{p(a_i^l | \tau_i^l, z_i^t)}{p(a_i^l | \tau_i^l)} \right) + \sum_{l=t}^{t+k-1} \log \left( \frac{p(o_i^{l+1} | \tau_i^l, z_i^t, a_i^l)}{p(o_i^{l+1} | \tau_i^l, a_i^l)} \right) \right],$$

where $p(a_i^l | \tau_i^l, z_i^t)$ is the probability of taking action $a_i^l$ given the trajectory $\tau_i^l$ and role representation $z_i^t$, and $p(o_i^{l+1} | \tau_i^l, z_i^t, a_i^l)$ is the probability of observing $o_i^{l+1}$ given the trajectory $\tau_i^l$, action $a_i^l$, and role representation $z_i^t$.

*Proof.* We leverage the state-action dynamics of the MDP to formulate this proof. We use the fact that the probability of the future expected trajectory $\tau_i^{t+1:t+k}$ is the probability of the concatenated trajectory $\tau_i^{t+k}$ conditioned on the observation-action history $\tau_i^t$, i.e. $p(\tau_i^{t+1:t+k}) = p(\tau_i^{t+k} | \tau_i^t)$.

The conditional probability can be expanded through the Markov chain in the MDP to obtain the probability of the trajectory in terms of the future actions and observations. We write $p(\tau_i^{t+k} | \tau_i^t) = \prod_{l=t}^{t+k-1} p(a_i^l | \tau_i^l) p(o_i^{l+1} | \tau_i^l, a_i^l)$. Likewise, the conditional probability of the future trajectory given the role embedding $z_i^t$ is $p(\tau_i^{t+k} | \tau_i^t, z_i^t) = \prod_{l=t}^{t+k-1} p(a_i^l | \tau_i^l, z_i^t) p(o_i^{l+1} | \tau_i^l, a_i^l, z_i^t)$. It follows that the MI between the future trajectories and the role representations can be written as:

$$
\begin{aligned}
I(\tau_i^{t+1:t+k}; z_i^t) &= \mathbb{E}_{\tau_i^{t+k}, z_i^t} \left[ \log \frac{p(\tau_i^{t+1:t+k} \mid z_i^t)}{p(\tau_i^{t+1:t+k})} \right] \\
&= \mathbb{E}_{\tau_i^{t+k}, z_i^t} \left[ \log \frac{p(\tau_i^{t+k} \mid \tau_i^t, z_i^t)}{p(\tau_i^{t+k} \mid \tau_i^t)} \right] \\
&= \mathbb{E}_{\tau_i^{t+k}, z_i^t} \left[ \log \frac{\prod_{l=t}^{t+k-1} p(a_i^l \mid \tau_i^l, z_i^t) p(o_i^{l+1} \mid \tau_i^l, a_i^l, z_i^t)}{\prod_{l=t}^{t+k-1} p(a_i^l \mid \tau_i^l) p(o_i^{l+1} \mid \tau_i^l, a_i^l)} \right] \\
&= \mathbb{E}_{\tau_i^{t+k}, z_i^t} \left[ \sum_{l=t}^{t+k-1} \left( \log \left( \frac{p(a_i^l \mid \tau_i^l, z_i^t)}{p(a_i^l \mid \tau_i^l)} \right) + \log \left( \frac{p(o_i^{l+1} \mid \tau_i^l, z_i^t, a_i^l)}{p(o_i^{l+1} \mid \tau_i^l, a_i^l)} \right) \right) \right]
\end{aligned}
$$

This concludes the proof. $\square$

# B. Implementation Details

### B.1. Contrastive Learning

We utilize the implementation of ACORM (Hu et al., 2024) to obtain the intermediate role embeddings necessary for the computation of the intrinsic rewards. First, the embeddings of the agent's observation-action history $e_i^t$ are periodically clustered into $|M|$ groups using K-means, where each $e_i^t$ is assigned to a group $C_{ij} = \{0, 1\}$, where $i \in I$ and $j \in \{0, ..., |M| - 1\}$. Using the cluster assignments $C$, *positive pairs* are generated from agent embeddings ($e_i^t$) and role representations ($z_i^t$) within the same cluster, while *negative pairs* are formed from embeddings across different clusters. More formally, for an agent $i$ in cluster $q$ ($C_{iq} = 1$), positive keys are $k^+ = \{f_{\theta_r'}(e_j^t) : j \in \oplus\}$, where $\oplus = [j : C_{jq} = 1]$, while negative keys are $k^- = \{f_{\theta_r'}(e_j^t) : j \notin \oplus\}$.

We use bilinear products (Laskin et al., 2020; Hu et al., 2024) to compute similarities between role embeddings and agent embeddings, respectively, to formulate a score function $g(z_i^t, e_j^t) = \exp(z_i^t W f_{\theta_r'}(e_j^t))$, where $W$ is a learnable parameter matrix. This score function measures the similarity between the query role representation $z_i^t$ of agent $i$ and a key $k \in k^+$ belonging to the same cluster. Finally, the overall loss function is given by

$$\mathcal{L} = -\log \frac{\sum_{k \in k^+} \exp(z_i^t W k)}{\sum_{k \in k^+} \exp(z_i^t W k) + \sum_{k \in k^-} \exp(z_i^t W k)}, \tag{14}$$

where the network parameter $\theta_r'$ is updated by MOCO (He et al., 2020) through network parameters $\theta_r$, i.e, $\theta_r' = (1 - \zeta)\theta_r + \zeta\theta_r'$, where $\zeta$ is the momentum hyperparameter.

### B.2. Networks

Our implementation details for the critic and role networks are similar to those in ACORM, while we utilize the implementation of DreamerV3 for the RSSM dynamics models to compute intrinsic rewards. We employ simple network architectures for the trajectory encoder, role encoder, and attention mechanism. The trajectory encoder consists of a fully connected multi-layer perceptron (MLP) and a GRU network with ReLU activation, which encodes an agent's trajectory into a 128-dimensional embedding vector. The role encoder is a fully connected MLP that converts the 128-dimensional trajectory embedding into a 64-dimensional role representation. For the mixing network, we follow the same configuration as (Rashid et al., 2020), which includes two hidden layers with 32 dimensions each and ReLU activation. Additionally, we follow the implementation of the mixing network in ACORM, which computes the attention weights between the embeddings of the global state trajectory and the individual role embedding representations. These attention weights are used to compute the mixing weights for all agents' utility functions in conjunction with the global state. For this mixing subnetwork, the dimension of the embedding of the global state trajectory is equal to the state embedding, which is 64. Further details of these network structures are summarized in Table 1.

The learning hyperparameters are consistent with those of ACORM, where we use the Adam optimizer with a learning rate of $6 \times 10^{-4}$. Exploration is conducted using an $\epsilon$-greedy strategy, where $\epsilon$ is linearly reduced from 1.0 to 0.02 over 80,000 timesteps and remains constant afterward. Episodes collected through online interactions are stored in a replay buffer with a capacity of 5,000 state transitions, and Q-networks are updated using batches of 32 episodes sampled from this buffer. The target Q-network is updated using a soft update strategy with a momentum coefficient of 0.005. Additionally, the contrastive learning loss is optimized jointly for every 100 Q-network updates. All the learned decentralized policies are evaluated every 5,000 updates using 32 generated episodes. For most runs, we set the number of clusters to 3. We detail all hyperparameters in Table 2, and we outline any deviations from the standard used hyperparameters within the following table.

*Table 1.* The network configurations used for ACORM based on QMIX

| Network Configurations | Value | Network Configurations | Value |
|---|---|---|---|
| role representation dim | 64 | hypernetwork hidden dim | 32 |
| agent embedding dim | 128 | hypernetwork layers num | 2 |
| state embedding dim | 64 | type of optimizer | Adam |
| attention output dim | 64 | activation function | ReLU |
| attention head num | 4 | add last action | True |
| attention embedding dim | 128 | | |

*Table 2.* Hyperparameters used for ACORM under SMAC and SMAC-V2.

| Hyperparameter | SMAC | SMAC-V2 |
|---|---|---|
| buffer size | 5000 | 5000 |
| batch size | 32, 64 for 6h_vs_8z | 32, 64 for protoss_10_vs_11, terran_10_vs_11, zerg_10_vs_11 |
| learning rate | $6 \times 10^{-4}$ | $6 \times 10^{-4}$ |
| use learning rate decay | True | True |
| contrastive learning rate | $8 \times 10^{-4}$ | $8 \times 10^{-4}$ |
| momentum coefficient $\beta$ | 0.005 | 0.005 |
| update contrastive loss interval $T_{cl}$ | 100 | 100 |
| start epsilon $\epsilon_s$ | 1.0 | 1.0 |
| finish epsilon $\epsilon_f$ | 0.02 | 0.02, 0.00 for zerg_5_vs_5, 0.001 for terran_10_vs_11 |
| $\epsilon$ decay steps | 80000 | 80000, 100000 for zerg_5_vs_5 |
| evaluate interval | 5000 | 5000 |
| evaluate times | 32 | 32 |
| target update interval | 200 | 200 |
| discount factor $\gamma$ | 0.99 | 0.99 |
| cluster num | 3, 5 for 5m_vs_6m | 3, 4 for terran_10_vs_11 |

## B.3. Intrinsic Reward

The complete intrinsic reward is given by the following equation:

$$
\begin{aligned}
r_{i,\text{int}}^t = & \beta_3 \left( \sum_{l=t}^{t+k-1} \mathbb{D}_{KL} \left( \text{SoftMax} \left( Q_i \left( \cdot \mid \tau_i^l, z_i^t; \phi_Q \right) \right) \| p \left( \cdot \mid \tau_i^l \right) \right) \right) \\
& + \sum_{l=t}^{t+k-1} \beta_1 \left( \log q_{\psi_{\text{dec}}} \left( o_i^{l+1} \mid h_i^{l+1}, d_i^{l+1}, z_i^t \right) + \beta_2 \log q_{\psi_{\text{dyn}}} \left( d_i^{l+1} \mid h_i^{l+1}, z_i^t \right) \right) \\
& - \sum_{l=t}^{t+k-1} \left( \log p_{\phi_{\text{dec}}} \left( o_i^{l+1} \mid h_i^{l+1}, d_i^{l+1} \right) + \beta_2 \log p_{\phi_{\text{dyn}}} \left( d_i^{l+1} \mid h_i^{l+1} \right) \right),
\end{aligned}
$$

where $q_{\psi_{\text{dyn}}}(d_i^t \mid h_i^t, z_i^t)$ with parameters $\psi_{\text{dyn}}$ is the role-conditioned latent dynamics predictor of the RSSM dynamics model in DreamerV3, $p_{\phi_{\text{dyn}}}(d_i^t \mid h_i^t)$ with parameters $\phi_{\text{dyn}}$ is the role-agnostic latent dynamics predictor of the RSSM model, $q_{\psi_{\text{dec}}}(o_i^t \mid h_i^t, d_i^t, z_i^t)$ is the role-conditioned dynamics decoder of the model with parameters $\psi_{\text{dec}}$, and $p_{\phi_{\text{dec}}}(o_i^t \mid h_i^t, d_i^t)$ is the role-agnostic dynamics decoder of the RSSM model with parameters $\phi_{\text{dec}}$. We detail the components and the training of both the models in the subsequent section. Additionally, $Q_i \left( \cdot \mid \tau_i^l, z_i^t; \phi_Q \right)$ are the agent's utility functions conditioned on the trajectory, and we set $p \left( \cdot \mid \tau_i^l \right)$ as

$$
p \left( \cdot \mid \tau_i^l \right) = \sum_{j \in I} \frac{1}{N} \text{SoftMax} \left( Q_i \left( \cdot \mid \tau_i^l, z_j^t; \phi_Q \right) \right), \tag{15}
$$

where $I$ is the set of $N$ agents. To simplify the hyperparameter tuning, we set $\beta_2$ to 0 and we set the decision horizon length $l$ to 1 to minimize training time. Note that the parameter $\beta_2$ controls the significance of the log-likelihood of the future latent state prediction given the current trajectory and role representation.

## B.4. DreamerV3 Implementation Details

We describe each of the architectural components and their corresponding network structures. The role-conditioned variant of the RSSM model utilized in Dreamer V3 (Hafner et al., 2023) comprises of the following components:

1. A sequence model $q_{\psi_{\text{seq}}}(h_i^t \mid \tau_i^{t-1})$ with parameters $\psi_{\text{seq}}$, encoding the trajectory $\tau_i^{t-1}$ into a hidden representation $h_i^t$.

*Table 3.* Intrinsic Reward Hyperparameters.

| Environment | Map | $\alpha$ | $\beta_1$ | $\beta_2$ | $\beta_3$ |
|---|---|---|---|---|---|
| SMAC | 5m_vs_6m | 0.05 | 2.0 | 0.0 | 1.0 |
| | MMM2 | 0.05 | 2.0 | 0.0 | 1.0 |
| | 3s5z_vs_3s6z | 0.1 | 1.0 | 0.0 | 1.0 |
| | 27m_vs_30m | 0.1 | 1.0 | 0.0 | 1.0 |
| | Corridor | 0.1 | 1.0 | 0.0 | 2.0 |
| | 6h_vs_8z | 0.10 | 0.9 | 0.0 | 2.0 |
| SMAC-V2 | protoss_5_vs_5 | 0.05 | 0.5 | 0.0 | 1.0 |
| | terran_5_vs_5 | 0.05 | 1.0 | 0.0 | 1.0 |
| | zerg_5_vs_5 | 0.05 | 0.2 | 0.0 | 0.5 |
| | protoss_10_vs_11 | 0.05 | 0.5 | 0.0 | 0.5 |
| | terran_10_vs_11 | 0.05 | 0.5 | 0.0 | 0.5 |
| | zerg_10_vs_11 | 0.05 | 0.2 | 0.0 | 1.0 |

*Table 4.* Network and Training Hyperparameters for Dreamer V3 Implementation

| Parameter | Value | Parameter | Value |
|---|---|---|---|
| Hidden Size | 128 | | 16 default |
| | | Batch Size | 32 for SMACv2 5_vs_5 suite |
| | | | 64 for SMACv2 10_vs_11 suite and SMAC 6h_vs_8z |
| Deterministic Hidden Size | 128 | Max Batch Length | Max Environment Steps |
| Dynamics Stochastic | 16 | Model Learning Rate | 1e-4 |
| Dynamics Latent Discretization | 16 | Dataset Buffer Size | 5000 |
| Encoder MLP Units | 128 | Initial Latent | 'Learned' |
| Encoder MLP Layers | 2 | Weight Decay | 0.0 |
| Decoder MLP Units | 128 | KL Nats | 512 |
| Decoder MLP Layers | 2 | Reconstruction Loss Scale | 1.0 |
| Activation | SiLU | Dynamics Loss Scale | 0.5 |
| Optimizer | Adam | Representation Loss Scale | 0.1 |
| Gradient Clip | 1000 | Latent Unimix Ratio | 0.01 |

The sequence model is implemented as an MLP that first maps the previous hidden state $h_i^{t-1}$, action, and observation $o_i^{t-1}$ to hidden features of size 128. This is subsequently passed to a GRU with recurrent depth 1 and a hidden embedding size of 128 to yield the hidden state features.

2. An observation encoder $q_{\psi_e}(d_i^t \mid h_i^t, o_i^t)$ with parameters $\psi_e$, mapping observations $o_i^t$ into a discrete latent representation $d_i^t$. The observation encoder comprises a 2-layer MLP with intermediate dimensions of size 128, to a 16-dimension latent vector where each dimension is discretized to 16 bins.

3. A dynamics predictor $q_{\psi_{\mathrm{dyn}}}(d_i^t \mid h_i^t, z_i^t)$ with parameters $\psi_{\mathrm{dyn}}$, predicting the latent representation $d_i^t$ from $h_i^t$. This is a single layer neural network.

4. An observation decoder $q_{\psi_{\mathrm{dec}}}(o_i^t \mid h_i^t, d_i^t, z_i^t)$ with parameters $\psi_{\mathrm{dec}}$, reconstructing observations $o_i^t$. This is 2-layer MLP network that maps the latent representation and hidden representation back to the observations. The intermediate dimension size here is 128.

We utilize the DreamerV3 torch implementation[1] and refer readers to their repository for more details on the implementation. Here, we describe the hyperparameters used in our training setup for the DreamerV3 RSSM model. We encourage the reader to refer to the DreamerV3 paper (Hafner et al., 2023) to understand the definitions of the hyperparameters.

---

[1] https://github.com/NM512/dreamerv3-torch

## B.5. Derivation of the Intrinsic Rewards

We obtain a lower bound for this ratio $\sum_{l=t}^{t+k-1} \log\left(\frac{p(a_i^l|\tau_i^l,z_i^t)}{p(a_i^l|\tau_i^l)}\right)$ in Eq. 5 by leveraging the non-negativity of the KL divergence $\mathbb{D}_{KL}\left(\text{SoftMax}\left(Q_i\left(\cdot \mid \tau_i^l, z_i^t; \phi_Q\right)\right) \mid\mid p\left(\cdot \mid \tau_i^l, z_i^t\right)\right)$. We outline how the lower bound is obtained:

$$\mathbb{E}_{z_i^t,\tau_i^l,a_i^l}\left[\log\left(\frac{p(a_i^l \mid \tau_i^l, z_i^t)}{p(a_i^l \mid \tau_i^l)}\right)\right] \geq \mathbb{E}_{z_i^t,\tau_i^l,a_i^l}\left[\log\left(\frac{\text{SoftMax}\left(Q_i\left(a_i^l \mid \tau_i^l, z_i^t; \phi_Q\right)\right)}{p(a_i^l \mid \tau_i^l)}\right)\right].$$

We begin with the ratio $\mathbb{E}_{z_i^t,\tau_i^l,a_i^l}\left[\log\left(\frac{p(a_i^l|\tau_i^l,z_i^t)}{p(a_i^l|\tau_i^l)}\right)\right]$ and write it as the following :

$$= \mathbb{E}_{z_i^t,\tau_i^l,a_i^l}\left[\log\left(\frac{p(a_i^l \mid \tau_i^l, z_i^t)}{\text{SoftMax}\left(Q_i\left(a_i^l \mid \tau_i^l, z_i^t; \phi_Q\right)\right)}\right) + \log\left(\frac{\text{SoftMax}\left(Q_i\left(a_i^l \mid \tau_i^l, z_i^t; \phi_Q\right)\right)}{p(a_i^l \mid \tau_i^l)}\right)\right]$$

$$= \mathbb{E}_{z_i^t,\tau_i^l}\left[\mathbb{E}_{a_i^t}\left[\log\left(\frac{\text{SoftMax}\left(Q_i\left(a_i^l \mid \tau_i^l, z_i^t; \phi_Q\right)\right)}{p(a_i^l \mid \tau_i^l)}\right)\right] - \mathbb{D}_{KL}\left(\text{SoftMax}\left(Q_i\left(\cdot \mid \tau_i^l, z_i^t; \phi_Q\right)\right) \mid\mid p\left(\cdot \mid \tau_i^l, z_i^t\right)\right)\right]$$

$$\geq \mathbb{E}_{z_i^t,\tau_i^l,a_i^l}\left[\log\left(\frac{\text{SoftMax}\left(Q_i\left(a_i^l \mid \tau_i^l, z_i^t; \phi_Q\right)\right)}{p(a_i^l \mid \tau_i^l)}\right)\right].$$

Therefore, we derive the intrinsic policy reward from the lower bound $\mathbb{E}_{z_i^t,\tau_i^l,a_i^l}\left[\log\left(\frac{\text{SoftMax}\left(Q_i\left(a_i^l|\tau_i^l,z_i^t;\phi_Q\right)\right)}{p(a_i^l|\tau_i^l)}\right)\right]$ and compute the intrinsic reward as the KL Divergence $\mathbb{D}_{KL}\left(\text{SoftMax}\left(Q_i\left(\cdot \mid \tau_i^l, z_i^t; \phi_Q\right)\right) \mid\mid p\left(\cdot \mid \tau_i^l\right)\right)$ between individual utilities $Q_i\left(\cdot \mid \tau_i^l, z_i^t; \phi_Q\right)$ and role agnostic action probability $p\left(\cdot \mid \tau_i^l\right)$.

Likewise, we derive the dynamics intrinsic reward to obtain the lower bound due to the non-negativity of the KL Divergence $\mathbb{D}_{KL}(q_\psi(o_i^{l+1} \mid \tau_i^l, z_i^t, a_i^l) \mid\mid p(o_i^{l+1} \mid \tau_i^l, z_i^t, a_i^l))$ as:

$$\mathbb{E}_{z_i^t,\tau_i^l,a_i^l}\left[\log\left(\frac{p(o_i^{l+1} \mid \tau_i^l, z_i^t, a_i^l)}{p(o_i^{l+1} \mid \tau_i^l, a_i^l)}\right)\right] \geq \mathbb{E}_{z_i^t,\tau_i^l,a_i^l}\left[\log\left(\frac{q_\psi(o_i^{l+1} \mid \tau_i^l, z_i^t, a_i^l)}{p(o_i^{l+1} \mid \tau_i^l, a_i^l)}\right)\right].$$

We can write the ratio $\mathbb{E}_{z_i^t,\tau_i^l,a_i^l}\left[\log\left(\frac{p(o_i^{l+1}|\tau_i^l,z_i^t,a_i^l)}{p(o_i^{l+1}|\tau_i^l,a_i^l)}\right)\right]$ as

$$= \mathbb{E}_{z_i^t,\tau_i^l,a_i^l}\left[\log\left(\frac{p(o_i^{l+1} \mid \tau_i^l, z_i^t, a_i^l)}{p(o_i^{l+1} \mid \tau_i^l, a_i^l)}\right)\right]$$

$$= \mathbb{E}_{z_i^t,\tau_i^l,a_i^l}\left[\log\left(\frac{p(o_i^{l+1} \mid \tau_i^l, z_i^t, a_i^l)}{q_\psi(o_i^{l+1} \mid \tau_i^l, z_i^t, a_i^l)}\right) + \log\left(\frac{q_\psi(o_i^{l+1} \mid \tau_i^l, z_i^t, a_i^l)}{p(o_i^{l+1} \mid \tau_i^l, a_i^l)}\right)\right]$$

$$= \mathbb{E}_{z_i^t,\tau_i^l,a_i^l}\left[\log\left(\frac{q_\psi(o_i^{l+1} \mid \tau_i^l, z_i^t, a_i^l)}{p(o_i^{l+1} \mid \tau_i^l, a_i^l)}\right) - \mathbb{D}_{KL}(q_\psi(o_i^{l+1} \mid \tau_i^l, z_i^t, a_i^l) \mid\mid p(o_i^{l+1} \mid \tau_i^l, z_i^t, a_i^l))\right]$$

$$\geq \mathbb{E}_{z_i^t,\tau_i^l,a_i^l}\left[\log\left(\frac{q_\psi(o_i^{l+1} \mid \tau_i^l, z_i^t, a_i^l)}{p(o_i^{l+1} \mid \tau_i^l, a_i^l)}\right)\right].$$

This yields the above-mentioned lower bound. From this, we derive the intrinsic rewards. Note that we use to separate models, a role conditioned DreamerV3 model $q_\psi(o_i^{l+1} \mid \tau_i^l, z_i^t, a_i^l)$ and a role-agnostic DreamerV3 model $p(o_i^{l+1} \mid \tau_i^l, a_i^l)$. Here, the RSSM model $q_\psi(o_i^{l+1} \mid \tau_i^l, z_i^t, a_i^l)$ can be written as $q_\psi(o_i^{l+1} \mid \tau_i^l, z_i^t, a_i^l) = q_{\psi_{\text{seq}}}(h_i^{t+1} \mid \tau_i^t)q_{\psi_{\text{dyn}}}(d_i^{t+1} \mid h_i^{t+1}, z_i^t)q_{\psi_{\text{dec}}}(o_i^t \mid h_i^{t+1}, d_i^{t+1}, z_i^t)$, and $p(o_i^{l+1} \mid \tau_i^l, a_i^l) = p_{\phi_{\text{seq}}}(h_i^{t+1} \mid \tau_i^t)p_{\phi_{\text{dyn}}}(d_i^{t+1} \mid h_i^{t+1})p_{\phi_{\text{dec}}}(o_i^t \mid h_i^{t+1}, d_i^{t+1})$.

Therefore, the ratio $\mathbb{E}_{z_i^t, \tau_i^l, a_i^l} \left[ \log \left( \frac{q_\psi(o_i^{l+1} \mid \tau_i^l, z_i^t, a_i^l)}{p(o_i^{l+1} \mid \tau_i^l, a_i^l)} \right) \right]$ is given as:

$$= \mathbb{E}_{z_i^t, \tau_i^l, a_i^l} \left[ \log \left( \frac{q_{\psi_{\text{seq}}}(h_i^{t+1} \mid \tau_i^t) q_{\psi_{\text{dyn}}}(d_i^{t+1} \mid h_i^{t+1}, z_i^t) q_{\psi_{\text{dec}}}(o_i^t \mid h_i^{t+1}, d_i^{t+1}, z_i^t)}{f_{\phi_{\text{seq}}}(h_i^{t+1} \mid \tau_i^t) p_{\phi_{\text{dyn}}}(d_i^{t+1} \mid h_i^{t+1}) p_{\phi_{\text{dec}}}(o_i^t \mid h_i^{t+1}, d_i^{t+1})} \right) \right]$$

$$\approx \mathbb{E}_{z_i^t, \tau_i^l, a_i^l} \left[ \log \left( \frac{q_{\psi_{\text{dyn}}}(d_i^{t+1} \mid h_i^{t+1}, z_i^t) q_{\psi_{\text{dec}}}(o_i^t \mid h_i^{t+1}, d_i^{t+1}, z_i^t)}{p_{\phi_{\text{dyn}}}(d_i^{t+1} \mid h_i^{t+1}) p_{\phi_{\text{dec}}}(o_i^t \mid h_i^{t+1}, d_i^{t+1})} \right) \right] \quad \text{(sequence models will yield similar hidden states)}$$

$$= \left( \log q_{\psi_{\text{dec}}}(o_i^{l+1} \mid h_i^{l+1}, d_i^{l+1}, z_i^t) + \log q_{\psi_{\text{dyn}}}(d_i^{l+1} \mid h_i^{l+1}, z_i^t) \right)$$
$$- \left( \log p_{\phi_{\text{dec}}}(o_i^{l+1} \mid h_i^{l+1}, d_i^{l+1}) + \log p_{\phi_{\text{dyn}}}(d_i^{l+1} \mid h_i^{l+1}) \right).$$

We add the intrinsic reward hyper-parameters $\beta_1$ to regulate the contributions of the log-likelihood of the observations under the role-conditioned model. Likewise, we add the hyperparameter $\beta_2$ to regulate the relative weight of the log-likelihoods between of the observations through the decoder model and the future latent state under the dynamics predictor. The final dynamics intrinsic reward $r_{i,l+1,\text{dyn}}^t$ computed at a future timestep $l+1$ from a given timestep $t$ is given by:

$$r_{i,l+1,\text{dyn}}^t = \beta_1 \left( \log q_{\psi_{\text{dec}}}(o_i^{l+1} \mid h_i^{l+1}, d_i^{l+1}, z_i^t) + \beta_2 \log q_{\psi_{\text{dyn}}}(d_i^{l+1} \mid h_i^{l+1}, z_i^t) \right)$$
$$- \left( \log p_{\phi_{\text{dec}}}(o_i^{l+1} \mid h_i^{l+1}, d_i^{l+1}) + \beta_2 \log p_{\phi_{\text{dyn}}}(d_i^{l+1} \mid h_i^{l+1}) \right).$$

## C. Algorithm

We outline the algorithm for R3DM based on ACORM and highlight the differences in Blue.

---

**Algorithm 1** R3DM Based on ACORM and QMIX

---

**Input:** $\theta_e$: agent's trajectory encoder, $\theta_r$: role encoder, $|M|$: number of clusters, $\psi_{\text{seq}}, \psi_e, \psi_{\text{dyn}}$, and $\psi_{\text{dec}}$: role-conditioned DreamerV3 RSSM network parameters for sequence model, encoder model, dynamics model, and decoder model respectively, $\phi_{\text{seq}}, \phi_e, \phi_{\text{dyn}}$, and $\phi_{\text{dec}}$: role-agnostic DreamerV3 RSSM counterparts, $T_{cl}$: time interval for updating contrastive loss, $n$: number of agents, $B$: replay buffer, $T$: time horizon of a learning episode, $\zeta$: MOCO momentum hyperparameter, $\theta'_r$: key role encoder, $\alpha, \beta_1, \beta_2, \beta_3$: Intrinsic Reward hyperparameters

**Output:** Parameters of individual Q-network $\phi_Q$ and mixing network $\phi$.

1: Initialize all network parameters and replay buffer $B$ for storing agent trajectories.
2: **for** episode = 1, 2, … **do**
3:     Initialize history agent embedding $e_i^0$ and action vector $a_i^0$ for each agent.
4:     **for** $t = 1, 2, \ldots, T$ **do**
5:         Obtain each agent's partial observation $\{o_i^t\}_{i=1}^n$ and global state $s^t$.
6:         **for** agent $i = 1, 2, \ldots, n$ **do**
7:             Calculate the agent embedding $e_i^t = f_{\theta_e}(o_i^t; a_i^{t-1}, e_i^{t-1})$. and the role representation $z_i^t = f_{\theta_r}(e_i^t)$.
8:             Select the local action $a_i^t$ according to the individual Q-function $Q_i(e_i^t, a_i^t)$.
9:         **end for**
10:       Execute joint action $\mathbf{a}^t = [a_1^t, a_2^t, \ldots, a_n^t]^\top$, and obtain global reward $r^t$.
11:     **end for**
12:     Append the complete trajectory to $B = B \bigcup \{[o_i^t]_{i=1}^n, s^t, [a_i^t]_{i=1}^n, r^t, [e_i^t]_{i=1}^n, [z_i^t]_{i=1}^n\}_{t=1}^T$ .
13:     **if** episode mod $T_{cl} == 0$ **then**
14:         Sample a batch of trajectories from $B$.
15:         Partition agent embeddings $\{e_i^t\}_{i=1}^n$ into $|M|$ clusters to obtain cluster allocation matrix $C_{ij} = \{0, 1\}$ where $j \in \{0, ..., |M| - 1\}$ using K-means.
16:         Update key role encoder parameters $\theta'_r = (1 - \zeta)\theta_r + \zeta\theta'_r$
17:         **for** agent $i = 1, 2, \ldots, n$ **do**
18:             Obtain cluster $q$ such that $C_{iq} = 1$
19:             Construct positive keys $k^+ = \{f_{\theta'_r}(e_j^t) : j \in \oplus\}$, where $\oplus = [j : C_{jq} = 1]$, while negative keys are $k^- = \{f_{\theta'_r}(e_j^t) : j \notin \oplus\}$
20:         **end for**
21:         Update contrastive learning loss according to Eq. 14.
22:     **end if**
23:     Sample minibatch $B' \in B$ and update RSSM network parameters $\psi_{\text{seq}}, \psi_e, \psi_{\text{dyn}}$, and $\psi_{\text{dec}}$.
24:     Sample minibatch $B'' \in B$ and update role-agnostic RSSM network parameters $\phi_{\text{seq}}, \phi_e, \phi_{\text{dyn}}$, and $\phi_{\text{dec}}$.
25:     Sample minibatch $B''' \in B$ of size $K$.
26:     **for** $k = 1, 2, \ldots, K$ **do**
27:         **for** time $t = 1, 2 \ldots, n$ **do**
28:             Set $r_{\text{int},k}^t = 0$.
29:             **for** agent $i = 1, 2, \ldots, n$ **do**
30:                 Compute $r_{i,k,t}^{\text{pol}}$ from Eq 7 from individual Q-network for $i$ and role representation $z_{i,k}^t$.
31:                 Compute $r_{i,k,t}^{\text{dyn}}$ from Eq 9 from $z_{i,k}^t$ through models with parameters $\psi_{\text{dyn}}, \psi_{\text{dec}}, \phi_{\text{dyn}}$, and $\phi_{\text{dec}}$.
32:             **end for**
33:             Set $r_{\text{int},k}^t = \sum_{i \in I} \beta_3 r_{i,kt}^{\text{pol}} + r_{i,k,t}^{\text{dyn}}$.
34:             Set $r_k^t = \alpha r_{\text{int},k}^t + r_k^t$.
35:         **end for**
36:     **end for**
37:     Update the parameters of individual Q-network $\phi_Q$ and the mixing network $\phi$ with modified minibatch $B'''$.
38: **end for**

---

