# OpenReview forum: "R3DM: Enabling Role Discovery and Diversity Through Dynamics Models in Multi-agent Reinforcement Learning"
_ICML.cc/2025/Conference — ICML 2025 poster_

### Official Review · Reviewer_3pvG · 2025-03-13

**Overall Recommendation:** 4

**Summary:**

This paper tackles the problem of high quality multi agent reinforcement learning. Specifically, allowing individual agents to learn unique policies in order to better collaborate to achieve goals. The authors introduce a new approach to training, R3DM, which utilizes contrast learning to encourage individual agents to learn different policies based on their history and other agents in their group. The method is tested on several problems and appears to outperform standard methods.

## Update after rebuttal
The initial review was positive and I did not feel the need to change it after the rebuttal.

**Claims And Evidence:**

The authors provide adequate evidence for each of their claims.

**Essential References Not Discussed:**

I do not believe any essential references were not discussed.

**Experimental Designs Or Analyses:**

The experiments are standard in the field and therefore sound.

**Methods And Evaluation Criteria:**

The methods appear to be sufficient and make sense for the problem being studied.

**Other Comments Or Suggestions:**

Outside of the questions posed, I do not have suggestions.

**Other Strengths And Weaknesses:**

The study appears quite complete. It is well written and I find the study to be innovative.

**Questions For Authors:**

1. Can you quantify the role the contrastive learning is playing? Does this need to be performed throughout all of the training or can it be done later once shared policies have been learned?
2. What is the cost increase of using the algorithm?
3. How diverse of a strategy can emerge from this algorithm?
4. Have the authors compared their approach against using a shared base policy with small single layers for individual agents.

**Relation To Broader Scientific Literature:**

The methods introduced are of great interest to the scientific community, specifically, encouraging diverse behavior in multi agent systems without introducing the huge overhead of individual models for each of them.

**Theoretical Claims:**

There are some theoretical claims made but they do not form the basis of the results.

---

> ### Author Rebuttal · Authors · 2025-04-01
>
> We thank the reviewer for their positive feedback on the paper and for raising interesting and relevant questions. We aim to highlight this clearly in the camera-ready version of the paper.
>
> 1) Impact and Details on Contrastive Learning
>
>  Contrastive learning plays a crucial role in R3DM by deriving intermediate role embeddings, essential for enabling role-specific intrinsic rewards that drive diverse future behaviors. Contrastive learning is conducted throughout the training process. Specifically, following the training protocol outlined in ACORM (Hu et al., 2024), contrastive learning updates occur after every 100 episodes of data collection. We refer the reviewer to the ablation study under the rebuttal for reviewer STTJ for more details.
>
> 2) Compute Cost Increase
>
> The proposed R3DM algorithm does not incur additional computational overhead during the testing or deployment phase, as only the learned policy is executed. The overhead introduced by R3DM arises solely during training, specifically due to the learning of an additional dynamics model required for computing intrinsic rewards. This extra step significantly enhances coordination through role-specific behavior. This extra step results in the increases moderately increases the training time of R3DM by 50-75% compared to the baselines. With more efficient implementations of the dynamics model on frameworks such as Jax, we believe that the training run-time can be optimized further.
>
> 3.  Diversity of Strategy
>
> Our approach inherently encourages strategic diversity as a mechanism to optimize coordination objectives, which in turn improves sample complexity and efficiency. Qualitative analyses provided in the paper illustrate that R3DM successfully facilitates the emergence of diverse and sophisticated strategies. For instance, agents have learned distinct tactics such as deliberately distracting subsets of enemy agents to weaken their overall strength, thereby enabling other teammates to effectively neutralize remaining threats.
>
> 4. Have the authors compared their approach against using a shared base policy with small single layers for individual agents?
>
> Yes. In our experiments, the baseline QMIX algorithm inherently implements a shared local critic structure that agents use to formulate policies for choosing the desired actions. This implementation persists in R3DM, where we use a shared local critic and the shared role embedding network across multiple agents. We provide the anonymized code here
> https://anonymous.4open.science/r/R3DM-F1A0/README.md.

---

> > ### Comment · Reviewer_3pvG · 2025-04-02
> >
> > I thank the authors for their detailed response. I don't feel I need to change the recommendation.

---

> > > ### Author Response · Authors · 2025-04-06
> > >
> > > Thank you very much for the opportunity to improve our work. We will revise the manuscript to address your questions.

---

### Official Review · Reviewer_J2zb · 2025-03-14

**Overall Recommendation:** 3

**Summary:**

The paper proposes a method for improving the ability of agents to learn to effectively coordinate. The method is based on an existing idea that clusters learning agents into roles based on their observation history using contrastive learning but extends this idea by encouraging diversity among the roles to aid exploration and role selection. The paper evaluates the method in SMAC and SMACv2 and compares against several baselines, finding that their method outperforms the baselines in sample complexity in some scenarios (but rarely underperforms).

**Claims And Evidence:**

Overall, the paper does a poor job of communicating what their contributions are, especially in relation to ACORN. Unless I've misunderstood, the contrastive learning objective and clustering strategy (sections 4.1 and 4.2) is identical to ACORN, but this is not made clear in the paper. If this is the case, using the terminology "building on" and "inspired by" is misleading on terms of the contribution of the paper and 4.1 and 4.2 should be phrased as preliminary or existing work. If the details in 4.1 and 4.2 do indeed differ from ACORN, these differences should be made clear.

The claims that R3DM improves in sample complexity over baselines is relatively convincing. The experiments are limited to the SMAC and closely related SMACv2 environments so the generalizability of the claim is not clear.

**Essential References Not Discussed:**

One of the key points made in the paper is that the diversity of trajectories across roles can help improve learning. Existing literature [1] has explored the problems with entropy-based diversity in coordination settings. It would be interesting to see a discussion with respect to this line of work.

[1] Cui, Brandon, et al. "Adversarial diversity in hanabi." The Eleventh International Conference on Learning Representations. 2023.

**Experimental Designs Or Analyses:**

All of the experimental claims are somewhat limited by the fact that only SMAC and SMACv2 are used as evaluation environments.

The experimental claims that R3DM is more sample efficient than existing algorithms is decently well supported although an improvement over ACORN is limited to only a few of the scenarios. However, it's not clear that R3DM converges to better solutions (rather than just more quickly) than other methods because training curves are cut off before convergence in most scenarios. It would be valuable to see experiments run to a larger number of environment steps.

**Methods And Evaluation Criteria:**

SMAC and SMACv2 are good domains for testing algorithm coordination abilities. A broader set of environments would substantially help strengthen the claims.

**Other Comments Or Suggestions:**

130 col. 2: "through [an] information-theoretic"
355 col. 2: spacing typo

**Other Strengths And Weaknesses:**

Strengths:

The paper proposes a new intrinsic reward that builds on top of the SOTA method for clustering-based coordination methods, ACORN. The idea is theoretically grounded and the experiments demonstrate the method improves the SOTA.

Weaknesses:

The experiment settings are limited. The improvements over baselines are quite small.

**Questions For Authors:**

What are the "test returns" recorded in SMACv2? (mentioned on line 353)

I notice that the clustering of roles changes throughout and the episode visualized in Figure 4. For example, the dead agents switch between cluster 1 and cluster 2 from step 35 to 50. Also, the number of agents in the other clusters changes over time. Why does this happen?

What are the error bars in the learning curve figures? Please specify this in the paper.

**Relation To Broader Scientific Literature:**

The key contribution of the paper is the extension of ACORN to include a new intrinsic reward to optimize role based diversity. The paper's evaluations show that this helps in the SMAC and SMACv2 environments.

**Theoretical Claims:**

I did not check the correctness of any proofs.

---

> ### Author Rebuttal · Authors · 2025-04-01
>
> We sincerely thank the reviewer for their time in providing detailed feedback on the paper to better highlight the key contributions and novelties of the paper. We detail our responses to the questions below.
>
> 1) Clarification of the contributions
>
> Our primary contribution lies in introducing a novel information-theoretic objective specifically tailored for role-based multi-agent reinforcement learning (MARL). We theoretically demonstrate that this objective can be naturally decomposed into two complementary sub-objectives. One that derives intermediate role embeddings from observation-action histories, which can be effectively optimized using contrastive learning, and
> another that refines these embeddings by aligning them more closely with future behaviors through intrinsic rewards.
> This decomposition ensures that roles derived from past interactions are not only compact representations of historical data but also predictive of future actions, thereby enabling role-specific coordination and specialization.
>
> While our framework incorporates contrastive learning and clustering techniques borrowed from ACORM, our approach subsumes this to optimize the broader optimized objective. Hence, they are integral to derive these intermediate role embeddings to optimize our proposed objective. For the sake of completeness and transparency, we have included them in the methodology section of our paper. To avoid any potential confusion, we will make this distinction clearer in the final camera ready version of the paper by explicitly identifying these elements as preliminary or existing work.
>
> 2)  Source Code
>
> We have uploaded and anonymized source code with the necessary data used for the plots in the paper and rebuttal (ablation studies for reviewer STTJ) to the best of our ability here: https://anonymous.4open.science/r/R3DM-F1A0/README.md
>
> 3) Clarification with respect to Entropy-based Methods
>
> [1] demonstrates that training a population of policies through adversarial diversity improves robustness by ensuring cross-policy compatibility. However, their approach requires maintaining multiple policies per agent to achieve competitive performance across permutations of partner policies, which increases complexity and computational overhead. We believe that this approach solves a related problem in the field of adhoc-teaming [2].  In contrast, our single-policy learning paradigm focuses exclusively on rapid task-reward convergence by leveraging role-driven trajectory diversity rather than optimizing for robustness across diverse partner policies. Entropy-based exploration methods such as CDS (which we benchmark against) are limited to static role assignments based on the identities of agents, which result in fixed behavior distributions that fail to adapt to evolving team dynamics. R3DM addresses this limitation by employing a mutual information objective (Theorem 4.1) to induce role emergence dynamically from observation-action histories. This allows agents to learn complementary roles based on the similarity or divergence of their trajectory histories. Additionally, R3DM’s intrinsic rewards are designed to capture diversity, specifically in future expected behaviors conditioned on the agent’s role, ensuring that roles evolve naturally and adaptively during interactions with the environment.
>
>
> 4) Test Returns
>
> The "test returns" mentioned in line 353 refer to cumulative rewards accumulated during testing episodes. We will clarify and correct this terminology in the final version of the paper to avoid ambiguity.
>
>
> 5) Changing of role-clusters
>
> Roles dynamically change throughout the episode because they depend on each individual agent’s evolving observation-action history. Thus, as agents interact with the environment and their experiences diverge over time, their corresponding roles naturally adjust, causing fluctuations in role cluster assignments, even for agents that become inactive or "dead.
>
> 6) Error Bars
>
> The error bars depicted in the learning curve figures represent the standard deviation computed across the 5 different random seeds used in our experiments. We will explicitly state this detail in the final manuscript to clarify the interpretation.
>
> [1] Cui, Brandon, et al. "Adversarial diversity in hanabi." The Eleventh International Conference on Learning Representations. 2023.
>
> [2] Mirsky, Reuth, et al. "A survey of ad hoc teamwork research." European conference on multi-agent systems. Cham: Springer International Publishing, 2022.

---

### Official Review · Reviewer_STTJ · 2025-03-14

**Overall Recommendation:** 3

**Summary:**

The authors propose R3DM, a new role-based MARL framework that enhances coordination by learning roles that shape agents' future behavior through maximizing mutual information and using intrinsic rewards derived from dynamics models.

**Claims And Evidence:**

The core claims regarding R3DM's ability to learn effective coordination and outperform baselines on SMAC and SMACv2 are strongly supported by clear and convincing evidence through quantitative results, qualitative analysis, and a theoretical framework. The acknowledged limitations suggest avenues for future improvement but do not undermine the validity of the findings.

**Essential References Not Discussed:**

N/A

**Experimental Designs Or Analyses:**

There is a lack of ablation studies, so it is unclear which components of the method are effective.

**Methods And Evaluation Criteria:**

Generally yes, but:
1.The authors could have also shown the effect of varying the number of roles.
2. There is a lack of ablation studies, so it is unclear which components of the method are effective.

**Other Comments Or Suggestions:**

1. An overall block diagram describing the method would have been useful.

2. Regarding the intrinsic reward - While the mathematical derivation is provided in the Appendix, the core idea of how such a reward encourages role-aligned diverse policies should be highlighted in the main text.

3. The qualitative results could be improved by visualising the learned policies.

4. Ablation studies showing the effect of each component of the proposed methods is needed.

**Other Strengths And Weaknesses:**

The paper is well motivated in an intuitive manner and presents a novel method grounded in an information theoretic objective. The results are impressive, but there is a lack of ablation studies and a lack of insight on how varying the number of roles affects performance.

**Questions For Authors:**

1. The motivating example is useful, but I was wondering - what if it is indeed better for both drones to target the same building (eg: 1 drone alone cannot put out the fire)? This points to a more general question - does the approach take into consideration the difficulty of the task under consideration? How would one balance task difficulty with diversity?

2. The MI objective could be explained in further detail-  why does this objective lead to the desired role properties?

3. Having a fixed number of roles is a limitation. However, while the paper shows good performance with a fixed number of roles, how would the performance vary with different numbers of predefined roles?

4. Having two dynamic models, one for role-conditioning and one for role-agnostic could be expensive. Have the authors thought about how a single dynamics model may be used?

**Relation To Broader Scientific Literature:**

The method leverages and extends existing ideas in role-based MARL and contrastive learning, while introducing a novel approach to intrinsic reward design based on predicting the impact of roles on future trajectories using a dynamics model. This forward-looking perspective on role learning seems to be the key contribution.

**Theoretical Claims:**

I did not check the proofs in the appendix in detail.

---

> ### Author Rebuttal · Authors · 2025-04-01
>
> We sincerely thank the reviewer for their detailed and thoughtful feedback, which has greatly contributed to improving the scientific rigour of this paper. We detail our responses to the questions below and add the required ablation studies.
>
> 1) Intrinsic Reward:
>
> We clarify the intuition behind our intrinsic rewards. The rewards encourage agents to diversify their future behaviors, conditioned explicitly on their assigned roles. Formally, it represents the difference between the entropy of future trajectories and the conditional entropy of these future trajectories given specific roles. Therefore, the intrinsic reward inherently promotes diversity by increasing the overall trajectory entropy, ensuring diverse future behaviors while simultaneously maintaining the coherence and alignment of behaviors with respect to roles.
>
> 2)  Clarification on the MI Objective :
>
> The MI objective in R3DM is designed to ensure that an agent’s current role influences its future behavior while being grounded in its prior observations. This objective quantifies the dependency between the concatenated observation-action trajectory (which includes both the agent’s observation-action history and its future trajectory) and its role. Intuitively, maximizing this MI objective ensures that the roles derived from observation-action histories can effectively predict future actions and observations. This, in turn, enables better specialization within the team, which facilitates more sample-efficient multi-agent learning, as demonstrated in our experiments.
>
> 3) Ablations:
>
> We conduct 3 ablations on the design choices. The plots are in the link.
> https://drive.google.com/file/d/1b4IkSXhNhXFedi2-d8K8tTEJjCVz4hey/view?usp=sharing
>
> a) Impact of the reward imagination: We analyze the impact of the number of imagination steps in the role-conditioned future trajectory, which is used to predict intrinsic rewards. In R3DM, we use a single imagination step to compute intrinsic rewards. We observe no statistically significant improvements when increasing the number of imagination steps to 2. However, with a higher number of imagination steps (e.g., 5 and 10), we observe a degradation in performance (more pronounced for 10 steps). We believe this degradation is due to compounding errors in the predictions by the model, which is conditioned solely on an agent’s past observations and actions. These errors lead to intrinsic rewards with higher variance and bias, ultimately degrading learning.
>
> b) Impact of the number of roles
>
> We vary the number of role clusters used in R3DM progressively from 2 to 8 for the 3s5z_vs_3s6z environment. We don’t observe a significant difference in final performances when varying the number of role clusters. We observe that the learning with 3 role clusters is more sample efficient compared to other role clusters.
>
> c) Impact of Contrastive Learning (CL)
>
> We compare the influence of intrinsic rewards and CL on the performance of R3DM. The variant of our R3DM without intrinsic rewards is equivalent to ACORM, as our method builds upon it. Next, we evaluate a variant of our algorithm that includes intrinsic rewards but excludes CL, which is used to enable more distinct role clusters. We observe that this degrades performance CL. Interestingly, this variant still outperforms ACORM, underscoring the significant impact of our proposed intrinsic rewards. These rewards enhance the entropy of future trajectories based on roles while reducing their conditional entropy with respect to intermediate role representations.
>
> 4) Balancing task rewards with exploration diversity
>
> The reviewer raises an insightful point regarding scenarios where it would be optimal for multiple agents (e.g., drones) to collaboratively target the same objective due to task difficulty. Our proposed approach accounts for such task-specific complexities by integrating intrinsic rewards with task-specific rewards. The balance between these two reward types is controlled by the hyperparameter \alpha, which can be tuned based on the requirements of the specific task. This design allows our method to flexibly adjust the trade-off between promoting diversity and fostering cooperation, depending on the inherent difficulty and coordination demands of the task.
>
> 5) Using a single dynamics model:
>
> The reviewer rightly points out that employing two separate dynamics models (one role-conditioned and one role-agnostic) increases training computational overhead. Importantly, this overhead is confined to the training phase, and dynamics models are not needed during the testing or inference phase. We have explored the possibility of using a single dynamics model with multiple forward passes using different sampled role embeddings to compute intrinsic rewards. However, empirically, we observed this approach to be computationally slower due to the necessity of multiple forward passes with different role embeddings when computing the intrinsic reward.

---

> > ### Comment · Reviewer_STTJ · 2025-04-08
> >
> > I thank the authors for their detailed responses. I have updated my score accordingly.

---

### Decision · Program_Chairs · 2025-05-01

**Decision:**

Accept (poster)

**Comment:**

This paper presents the idea that an agent's role should influence its future actions for effective coordination and then proposes R3DM, a new role-based MARL framework. R3DM learns emergent roles by maximizing mutual information among agents' roles, observed paths, and expected future behaviors, optimizing the objective via contrastive learning on past trajectories to get intermediate roles which shape rewards to boost future behavioral diversity across roles using a learned dynamics model. Overall, this paper is interesting and with solid technical contributions to MARL. Thus I tend to accept this paper.